# LeapFactual: Reliable Visual Counterfactual Explanation Using Conditional Flow Matching

**Zhuo Cao**[1]    **Xuan Zhao**[1*]    **Lena Krieger**[1,2*]    **Hanno Scharr**[1]    **Ira Assent**[1,3]

[1]IAS-8, Forschungszentrum Jülich, Germany
[2]Munich Center for Machine Learning (MCML), LMU Munich, Germany
[3]Aarhus University, Denmark
`{z.cao, xu.zhao, l.krieger, h.scharr, i.assent}@fz-juelich.de`

## Abstract

The growing integration of machine learning (ML) and artificial intelligence (AI) models into high-stakes domains such as healthcare and scientific research calls for models that are not only accurate but also interpretable. Among the existing explainable methods, counterfactual explanations offer interpretability by identifying minimal changes to inputs that would alter a model's prediction, thus providing deeper insights. However, current counterfactual generation methods suffer from critical limitations, including gradient vanishing, discontinuous latent spaces, and an overreliance on the alignment between learned and true decision boundaries. To overcome these limitations, we propose LEAPFACTUAL, a novel counterfactual explanation algorithm based on conditional flow matching. LEAPFACTUAL generates reliable and informative counterfactuals, even when true and learned decision boundaries diverge. Following a model-agnostic approach, LEAPFACTUAL is not limited to models with differentiable loss functions. It can even handle human-in-the-loop systems, expanding the scope of counterfactual explanations to domains that require the participation of human annotators, such as citizen science. We provide extensive experiments on benchmark and real-world datasets highlighting that LEAPFACTUAL generates accurate and in-distribution counterfactual explanations that offer actionable insights. We observe, for instance, that our reliable counterfactual samples with labels aligning to ground truth can be beneficially used as new training data to enhance the model. The proposed method is diversely applicable and enhances scientific knowledge discovery as well as non-expert interpretability. The code is available on `https://github.com/caicairay/LeapFactual`.

## 1 Introduction

The widespread adoption of machine learning (ML) and Artificial Intelligence (AI) models in high-stakes domains — such as healthcare [1–7] or scientific research [8–10] — demands not only high performance but also transparency, reliability, and interpretability. In the context of scientific discovery, where reproducibility, verifiability, and a deep understanding of underlying mechanisms are paramount, opaque "black-box" models can hinder progress and erode trust in computational findings. Without interpretability, the outputs of ML models remain inscrutable, limiting their utility for hypothesis generation, experimental validation, and the advancement of scientific knowledge. In response, a variety of explainability methods have emerged to illuminate the inner workings of black-box models. Among the most common are gradient-based methods [11] and perturbation-based techniques [12, 13]. These methods generate a map of the regions that contribute most to the decision by either taking the backpropagation through the neural network or perturbing the input data. While

---

*These authors contributed equally.

39th Conference on Neural Information Processing Systems (NeurIPS 2025).

these methods offer insights on the location of the class-related features, they do not expose residual features (e.g., pattern in the location) or guide practitioners toward meaningful model refinements.

Counterfactual explanation (CE) has recently gained attention as a complementary, more informative alternative. By answering *'What would have needed to be different for the outcome to be different?'*, counterfactuals expose decision boundaries more clearly and can also be used to generate new, informative training samples that improve generalization and fairness [3, 7]. Despite their promise, existing counterfactual generation algorithms face significant limitations. Many struggle with gradient vanishing [14–18] or discontinuous latent spaces [19–24], leading to uninformative or unrealistic counterfactuals (see Sections 2.1 and 3.1). Moreover, CEs are typically located near the decision boundary. While this aligns with the definition of a counterfactual, it often fails to lie within the data distribution, thereby failing to accurately represent it (see Figure 1). Our algorithm generates counterfactual samples that remain within the distribution, a characteristic we refer to as reliability.

Addressing the aforementioned limitations is essential to realize the full potential of CEs for interpretability and model enhancement.

To address these challenges, our main contributions are:

- We introduce LEAPFACTUAL, a novel, reliable CE algorithm based on **conditional flow matching**. It retains the strengths of existing approaches while overcoming several critical shortcomings and handles scenarios where the true decision boundary deviates from the learned model.
- We propose the CE-CFM training objective, and provide **theoretical motivation** showing that flow matching naturally disentangles the class-related information from residual information. This clarifies how important features are isolated for counterfactual generation.
- We validate the effectiveness of LEAPFACTUAL on diverse benchmark datasets from different domains, including **astrophysics**, to demonstrate its applicability and performance across disciplines.
- We generate counterfactuals using **non-differentiable** models, such as **human annotators**, to illustrate that LEAPFACTUAL is model-agnostic, which significantly expands the applicability of CE to fields such as citizen science.

## 2  Preliminary and Related Work

We denote random variables with capital letters (e.g., $X$, $Y$, $Z$) and sets with script letters (e.g., $\mathcal{X}$, $\mathcal{Y}$, $\mathcal{Z}$). The realizations of variables are shown with lowercase letters (e.g. $x$, $y$, $z$). The probability density function of a distribution $\mathbb{P}(X)$ is represented by $p(x)$ or $q(x)$. The expectation with respect to $p(x)$ is denoted as $\mathbb{E}_{p(x)}[\cdot]$. We use lowercase letters, such as $f$, $g$, and $l$, to represent functions. The function of a neural network with parameters $\theta$ is represented as $f_\theta$. Loss functions are denoted as $\mathcal{L}$.

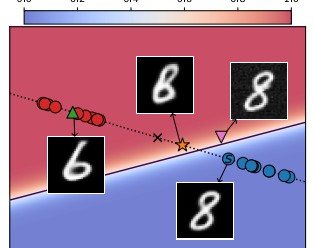 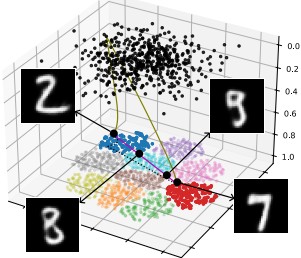

Figure 1: (**Left**) Conceptual illustration of 1D data embedded in 2D space. Blue ● and red ● circles denote data of two MNIST classes (8 and 6, resp.), with the dotted line representing the data manifold. The true decision boundary is ✕, while the solid line is the learned boundary. The background color indicates the output of the binary classifier. Adversarial attack ▽, CGM-generated CE △, and Opt-based CE ★ examples originate from the starting point ⑤. (**Right**) Conceptual multi-class 3D example. Each color represents an MNIST digit class. A sample from the blue cluster is transported along the olive path to reach the red cluster, forming a CE. The purple line shows the large distance between CE and the source, highlighting gradient vanishing issues. More details can be found in Section 3.1.

### 2.1  Counterfactual Explanation (CE)

Counterfactual explanations (CEs) [25, 26] modify an input data point in a semantically meaningful way to produce a similar counterpart with a different target label, offering deeper insight into model decisions. Unlike other explainability methods, CEs not only emphasize important features, but also provide actionable changes [3, 7], making them especially useful in

domains requiring transparency. Recently, generative models are the standard paradigm to produce in-distribution counterfactuals and avoid adversarial artifacts (see Section 3). There are two paradigms depending on the usage of generative models: Optimization (Opt)-based methods and conditional generative model (CGM)-based methods (see Figure 2).

Formally, given a classifier $f_\theta : \mathbb{R}^d \to [0,1]^n$ and an input $x \sim \mathbb{P}(X)$, the general objective for an Opt-based method [14–18] is to find a counterfactual $x_{\mathrm{CE}}$ by minimizing: $\mathcal{L}_{\mathrm{CE}}(x_{\mathrm{CE}}) = \mathcal{L}_{\mathrm{CLS}}(f_\theta(x_{\mathrm{CE}}), \hat{y}_c) + \lambda \mathcal{L}_{\mathrm{DIS}}(x, x_{\mathrm{CE}})$, where $\mathcal{L}_{\mathrm{CLS}}$ encourages the classifier $f_\theta$ to predict the target label $\hat{y}_c$ and $\mathcal{L}_{\mathrm{DIS}}$ enforces similarity to $x$, with $\lambda$ balancing the two terms. Note that overemphasis on similarity can yield adversarial examples ([27–29]) since the data ambient dimension is usually higher than the intrinsic dimension [30, 31] (see also Figure 1). A common solution is to prepend a generative model $g_\phi$ (e.g., VAE [32], GAN [33]) to $f_\theta$ to ensure the modification is in-distribution. This reformulates the loss to:

$$\mathcal{L}_{\mathrm{CE}}(z_{\mathrm{CE}}) = \mathcal{L}_{\mathrm{CLS}}(f_\theta(g_\phi(z_{\mathrm{CE}})), \hat{y}_c) + \lambda \mathcal{L}_{\mathrm{DIS}}(g_\phi(z), g_\phi(z_{\mathrm{CE}})), \tag{1}$$

where $z$ is the latent vector corresponding to the input $x$ from a VAE encoder or GAN inversion.

In contrast, CGM-based methods [19–24] integrate the information encoded in the classifier directly into the generative model, by training it conditioned on the output of the classifier.

Among various data modalities, generating CEs for visual data remains particularly challenging due to the high input dimension. Note, while our method focuses on visual data, it can be generalized to other data types. In the following, we summarize related work on visual counterfactual explanations regarding Opt-based and CGM-based methods.

**Optimization (Opt)-based Methods**
Opt-based methods prepend a generative model before the classifier to be studied, and optimize the latent vector corresponding to the input image towards the target label to generate CEs. REVISE [14] is a gradient-based method that samples from a generative model's latent space to find minimal changes altering predictions. Instead of directly optimizing latent vectors, Goetschalckx et al. [15] learn latent directions by differentiating through both generator and classifier. Other works consider training linear Support Vector Machines in latent space to control facial attributes [16] or using a GAN for image editing, guided by gradients to generate target-class images with minimal changes [17]. Dombrowski et al. [18] propose a theoretically grounded approach optimizing in the latent space of Normalizing Flows.

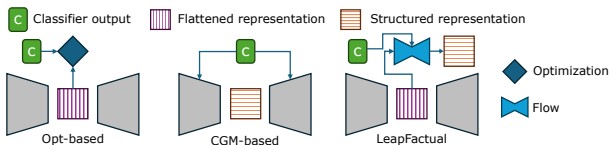

Figure 2: Overview of architectures - Opt-based models, CGM-based models and LEAPFACTUAL. Trapezoids represent generative models. Opt-based and CGM-based methods employ only flattened or structured representation (see Section 3.1), while LEAPFACTUAL combines both.

**Conditional Generative Model (CGM)-based Methods**
CGM-based methods incorporate the output of the classifier as a condition of the generative model, by replacing the condition counterfactual samples are generated. Samangouei et al. [19] jointly train classifier-specific encoders with a GAN to produce reconstructions, counterfactuals, and modification masks. To ensure realisitic counterfactuals, Singla et al. [20] use a GAN conditioned on classifier predictions with an encoder. DCEVAE [22] generates counterfactuals conditioned on input and target label. Other directions include training StyleGAN with classifier-specific style spaces for counterfactual manipulation [23], employing Normalizing Flows with a Gaussian mixture latent space and an information bottleneck to disentangle class information [24], and generating residuals added to the input to flip classifier decisions [21].

Table 1: Key characteristics of proposed method and baseline methods.

|  | Opt | CGM | Ours |
|---|---|---|---|
| Continuous latent space | ✓ | ✗ | ✓ |
| No gradient vanishing | ✗ | ✓ | ✓ |
| Pre-trained generator | ✓ | ✗ | ✓ |
| Model-agnostic | ✗ | ✓ | ✓ |
| Reliable CE | ✗ | ✗ | ✓ |

Unfortunately, both paradigms face significant drawbacks: Opt-based approaches enable continuous latent space exploration but face gradient vanishing and require differentiable classifiers; CGM-based methods mitigate vanishing gradients and offer structured representations but lack continuous latent spaces and often need separate generative models per classifier. The characteristics are summarized

in Table 1. The proposed method LEAPFACTUAL overcomes these issues and considers the mismatch between the learned and true decision boundaries to generate reliable CEs (see Section 3.1).

## 2.2 Flow Matching

The development of flow-based generative models dates back to Rezende et al. 2015 [34], which introduced Normalizing Flows—an invertible and efficiently differentiable mapping between a fixed distribution and the data distribution. Initially, these mappings were constructed as static compositions of invertible modules. Later, continuous normalizing flows (CNFs) were introduced, leveraging neural ordinary differential equations (ODEs) to model these transformations dynamically [35]. However, CNFs face significant challenges in training and scalability, particularly when applied to large datasets [35–37]. More recently, many works [38–42] demonstrated that CNFs could be trained using an alternative approach: regressing ODE's drift function, similar to how diffusion models are trained. This method, known as flow matching (FM), has been shown to improve sample quality and stabilize CNF training [38]. Initially, FM assumed a Gaussian source distribution, but subsequent generalizations have extended its applicability to more complex manifolds [43], arbitrary source distributions [44], and couplings between source and target samples derived from input data or inferred via optimal transport. In this work, we build upon I-CFM, the theoretical framework established by Tong et al. [42].

Formally, flow matching [38–42] learns a time-dependent vector field $u_t(z)$ whose flow pushes a source distribution $p_0$ to a target $p_1$. The dynamics

$$\mathrm{d}z = u_t(z)\,\mathrm{d}t$$

induce a probability path $p_t$ with $p_{t=0} = p_0$ and $p_{t=1} = p_1$. In *conditional flow matching (CFM)*, $p_t(z)$ is represented as a mixture over conditional paths $p_t(z \mid h)$ with $h := (z_0, z_1)$ [42]. Under *independent coupling*, $q(h) = q(z_0)q(z_1)$, the conditional path and target vector field are

$$\begin{aligned}
p_t(z \mid h) &= \mathcal{N}\big(z \,\big|\, (1-t)z_0 + tz_1,\, \sigma^2 I\big), \\
u_t(z \mid h) &= z_1 - z_0,
\end{aligned} \tag{2}$$

which defines I-CFM via a Gaussian flow with fixed variance $\sigma^2 I$.

In practice, a neural network $v_\psi(t, z)$ is trained to regress the target field $u_t(z \mid h)$ specified in Equation (2). Concretely, we sample

$$t \sim \mathrm{Unif}[0,1], \quad (z_0, z_1) \sim q(z_0)q(z_1), \quad z \sim p_t(\cdot \mid h),$$

and minimize a regression loss so that $v_\psi(t, z) \approx u_t(z \mid h)$. In I-CFM the target simplifies to the constant vector $z_1 - z_0$, independent of $t$ and $z$; the conditioning $h$ and the sampling of $z$ ensure the model learns the correct field along the path. At inference, integrating the learned field produces the flow map that transports samples from $p_0$ to $p_1$. Additional training details are provided in Section 3.2.1.

## 3 LEAPFACTUAL: A Counterfactual Explanation Algorithm

We motivate our approach by analysing the latent space of generative models in Section 3.1. We then demonstrate the flexibility of CFM in Section 3.2 and introduce our new algorithm LEAPFACTUAL.

### 3.1 Analysis of Latent Spaces

It is important to distinguish the latent representations used in unconditional generative models, employed by Opt-based methods, and conditional generative models, used in CGM-based methods. Let us assume a latent vector $z$ that is defined to be the result of a mapping function $m$, that maps class-related ($C$) and residual features ($R$) to the latent space: $z = m(c, r)$. For instance, in the MNIST dataset, $C$ denotes the digit identity, while $R$ captures the handwriting style. Ideally, a CE modifies only $C$, resulting in $z_{\mathrm{CE}} = m(c_{\mathrm{CE}}, r)$. In unconditional generative models, the latent space is *flattened* (Figure 3, left), meaning both $C$ and $R$ are entangled in $Z$. As a result, modifying $z$ inevitably affects both components, making it difficult to isolate class-related changes. Additionally, such models are trained to densely populate the latent space to cover the full data distribution

(Figure 1, right), which increases optimization complexity. Conversely, CGMs employ *structured* representations (Figure 3, middle), where class-related features $C$ are provided as external conditions, and the latent vector $Z$ encodes only the residual information $R$. In this case, $z = z_{CE}$, enabling CE generation by simply altering $c$ while keeping $r$ fixed. However, this structure leads to a *discontinuous* latent space, i.e., latent spaces of different classes are separated. This makes interpolation between samples non-trivial.

These limitations are illustrated in Figure 1. On the left, we depict data points on a 1D intrinsic manifold embedded in a 2D space. Generative models constrain CE samples to remain in-distribution, thereby avoiding adversarial artifacts ▽. In Opt-based methods, CEs ⭐ are obtained by optimizing the objective in Section 2.1. Typically, the optimization stops after crossing the learned decision boundary, without reaching the true decision boundary ✕. This results in uninformative and potentially misleading CEs, e.g., in the MNIST case, a CE classified as '6' may fail to exhibit key features of a genuine '6'.

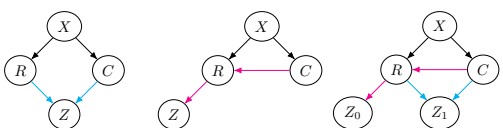

Figure 3: (**Left**) Flattened latent representation, (**Middle**) structured latent representation, and (**Right**) proposed latent representation, where $X$, $Z$, $C$, and $R$ represent input, the corresponding latent variable, class-related information, and residual information. The distributions connected by flow matching are $Z_0$ and $Z_1$.

More generally speaking, CEs close to the learned boundary are often not typical examples of their class. Further, samples between the learned and true decision boundaries are critical for improving model performance because they provide insights into the class features, which are unexplored under the Opt-based paradigm. This problem becomes more severe in multi-class scenarios (Figure 1, right), where the latent space is fully occupied and CEs may be far from the original input and pass through regions corresponding to other classes, making the gradient vanishing problem more significant. As a result, most Opt-based approaches are either limited to binary classification or require decomposition into binary subproblems. CGM-based methods alleviate some of these issues through structured and disentangled representations. However, they do not guarantee that the generated CE ▲ is close to the actual decision boundary. Moreover, their discontinuous latent spaces make reversing from the CE sample towards the decision boundary infeasible.

In addition, both Opt-based and CGM-based methods have practical limitations. Optimization-based approaches require a differentiable loss and are vulnerable to gradient decay in the generative model or classifier. CGM-based methods, on the other hand, must retrain the generative model for each new classifier — a costly and complex process — and demand careful architectural design [45] to ensure that conditioning information is preserved during training.

LEAPFACTUAL combines the advantages of both paradigms while mitigating the drawbacks. To achieve this, we propose a mapping between flattened and structured latent spaces by introducing a new latent dimension (Figures 1 to 3 right). Progressing along this axis removes class-related information, while reversing the direction re-injects it. Below, we present the algorithm in detail.

### 3.2 LeapFactual

In Section 3.2.1, we introduce the CE-CFM training objective and its theoretical justification for bridging the flattened and structured representations. Then, in Section 3.2.2, we show how a flow matching model trained with the CE-CFM objective can generate high-quality and reliable CEs.

### 3.2.1 Training Phase

We aim to leverage flow matching to bridge the gap between flattened and structured latent representations, as illustrated in the left and middle columns of Figures 2 and 3. However, this integration is not straightforward. As shown in the right column of the figures, our approach introduces a flow between $Z_0$ (structured encoding) and $Z_1$ (flattened encoding), mediated by their shared parent $R$. This design violates the independent coupling assumption inherent to the I-CFM framework, necessitating a modification of the original formulation. Given an input image $X$, a classifier $f_\theta$ extracts class-relevant information $C$, from which the residual $R$ can be subtracted. In MNIST digit classification, $C$ captures the digit identity, while $R$ corresponds to the writing style. Once $R$ is inferred from samples drawn from the data distribution $X$ and their corresponding class predictions

$C$, the latent variables $Z_0$ and $Z_1$ become *d-separated* [46]; i.e., they are conditionally independent given their only common parent $R$.

To address this, we redefine the conditioning term in I-CFM to $h := (z_0, z_{1,c})$. Hereby, $z_{1,c} \sim q(z_1 \mid c)$ denotes the latent vectors specific to the class $c = f_\theta(x)$ being predicted by the model, $z_0 \sim q(z_0)$ is randomly sampled from a Gaussian distribution.

The joint distribution can be expressed as $q(h) = q(z_0)q(z_1 \mid c)$ (derivation in Appendix A.1). To construct the probability path $p_t(z \mid h)$ and the corresponding vector field $u_t(z \mid h)$ as outlined in Equation 2, we sample from $q(z_0)$ and $q(z_1 \mid c)$. The samples are then inserted into the equation, thereby defining a transition from a Gaussian prior $q(z_0)$ to the class-specific distribution $q(z_1 \mid c)$. To generalize across classes, we condition the network on the predicted label $c = f_\theta(x)$, leading to our Counterfactual Explanation - Conditional Flow Matching (CE-CFM) objective:

$$\mathcal{L}_{\text{CE-CFM}}(\psi) := \mathbb{E}_{t, q(h), p_t(z|h)} \left\| v_\psi(t, z, c) - u_t(z \mid h) \right\|^2, \tag{3}$$

where $v_\psi$ is a neural network parameterized by $\psi$. Unlike the original I-CFM formulation [42], the CE-CFM objective explicitly incorporates the classifier output $c$ into $v_\psi$. Architecturally, this allows a single network to represent multiple class-specific flows. From an information-theoretic perspective, the Gaussian nature of $Z_0 \sim q(z_0)$ acts as an information bottleneck, compressing the content of $Z_1$. By including class information $C$ as a condition, the network can more efficiently discard class-related features during compression:

> **Theorem 1.** *Let $Z_1$ denote the original representation and $Z_0$ its counterpart obtained via flow matching conditioned on class information $C$. Then, $Z_0$ is a compressed representation of $Z_1$, and the optimized information loss incurred through this transformation corresponds exactly to the class-related information $C$ that is provided as a condition.*
>
> *Proof.* in Appendix A.2. □

Once the model is trained, $Z_0$ and $Z_1$ are linked through invertible transformations: a backward integral $\int_{t=1}^{0} v_\psi(t, z, c)\, dt$ and a forward integral $\int_{t=0}^{1} v_\psi(t, z, c)\, dt$. The backward pass effectively removes class-relevant information (compression), while the forward pass reintroduces it (reconstruction). For conceptual clarity, we refer to these two operations as the *lifting* and *landing* transports.

### 3.2.2 Explaining Phase

In principle, any generative model can be used to realize the proposed mapping—for example, a VAE encoder/decoder pair, GANs with inversion, or normalizing flows, which are inherently invertible. However, with flow matching, we achieve significantly greater flexibility, enabling not only information replacement but also more nuanced operations such as blending and injection.

We visualize the transport processes in Figure 4. Here, the latent space position is defined as $z = c_p + r_p$, where $c_p \in \{\pm 0.25\}^2$ denotes the four *center positions*, and $r_p \in (-0.25, 0.25)^2$ denotes the *relative position*. The relative position is constrained so that $z$ lies within the squares around the centers.

**Information Replacement**   It can be observed from Figure 4 (a) that multiple points from different classes in $Z_1$ are mapped to the same black point in $Z_0$, indicating that $Z_0$ is a compressed representation of $Z_1$, which complies Theorem 1. It is also evident in Figure 4 (b), where we fix the relative position $r_p$ while varying the center positions $c_p$, yielding the source points $z_{1,\text{blue}}$ ◯, $z_{1,\text{yellow}}$ ◯, and $z_{1,\text{green}}$ ◯. During the lifting transport, center information is discarded, resulting in similar transported points: $z_{0,\text{blue}} \approx z_{0,\text{yellow}} \approx z_{0,\text{green}}$. Consequently, the center information of the source points is fully replaced by the target label $c_{\text{red}}$ during the landing transport, and all points ●●● map to locations around $z_{1,\text{red}}$ ◯. This process produces a *global counterfactual* [47], where counterfactuals from different classes converge to the same result when their intra-class (residual) information is the same and targeting the same class, failing to explore inter-class counterfactuals.

**Information Blending**   For effective model explanations, *local counterfactuals* must preserve both intra- and inter-class relationships. Instead of fully replacing class-related features, meaningful counterfactuals blend source (with subscript $s$) and target (with subscript $t$) features. As shown in Figure 4 (c), blended points ●●● interpolate between source and target: $z = \alpha c_s + (1 - \alpha)c_t + r$,

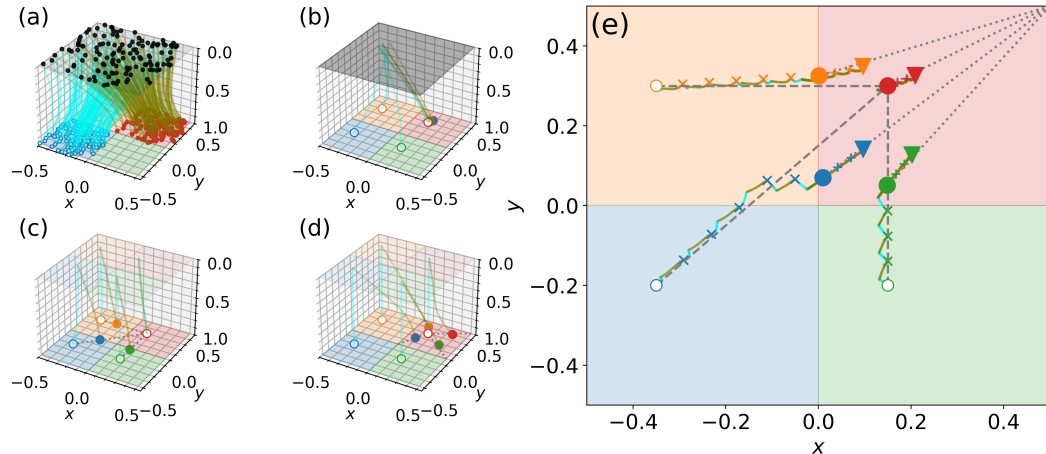

Figure 4: Toy experiments illustrating lifting and landing transports. The vertical axis represents $t$. Hollow circles denote source points; filled circles or triangles indicate transported points. **(a)** Blue source points are lifted to $t = 0$, removing *center position* information, which is replaced by the red class during landing. **(b)** Center information from green, blue, and yellow sources is overwritten by the red class. **(c)** Transported points blend center information from source (green, blue, red) and target (red); dashed lines show interpolation. **(d)** Target information is injected even when source and target are both red; dashed lines show interpolation. **(e)** At $t = 1$: filled circles mark blended points, triangles mark injected ones; dashed and dotted lines denote blending and injection interpolations.

with $\alpha < 1$. This is achieved by adjusting the transport step size $\gamma$ in $\int \gamma u_t \, dt$: $\gamma = 1$ yields full replacement, while $\gamma < 1$ enables blending. Blending can be applied progressively, and the classifier $f_\theta$ can track the class label at each step. This is useful in multi-class tasks where class identity may change mid-transport. For instance, in Figure 4 (e), the blue point transitions through the yellow region before reaching the target, deviating from the dashed interpolation path.

**Information Injection**   Stopping transport when the classifier prediction matches the target label may be insufficient, as the learned boundary may not reflect the true one (Section 3.1). To refine the result, target class information must be explicitly injected. Conditional lifting removes source class information, which cancels the effect of injection when source and target classes coincide (see red points ○● in Figure 4 (b), (c)). To prevent cancellation, we use a smaller step size for the lifting than for the landing transport. As shown in Figure 4 (d), where $\gamma_{\text{lift}} = 0$, $\gamma_{\text{land}} = 1$, the target red point ● follows: $z = (1 + \alpha)c_t + (1 - \alpha)r$, amplifying target class information when $c_s = c_t = c_{\text{red}}$. The relative position $r$ is also adjusted to maintain in-distribution samples. As shown in Figure 4 (e), the progressive injection after blending maximizes target class information.

---

**Algorithm 1** LEAP

**Input:**   Flow matching model $v_\psi$ trained with CE-CFM objective, source point $z_{\text{source}}$ (at $t=1$), current label $y_c$, target label $\hat{y}_c$, step sizes $\gamma_{\text{lift}}, \gamma_{\text{land}}$

**Step 1:**   Lift                                                                                         ▷ From $Z_1$ to $Z_0$

$$z^{y_c}(t) \;=\; z_{\text{source}} \;+\; \int_1^t \gamma_{\text{lift}} v_\psi\big(\tau,\, z^{y_c}(\tau),\, y_c\big)\, d\tau, \quad t \in [0, 1].$$

$z_{\text{lift}} \leftarrow z^{y_c}(0)$

**Step 2:**   Land                                                                                         ▷ From $Z_0$ to $Z_1$

$$z^{\hat{y}_c}(t) \;=\; z_{\text{lift}} \;+\; \int_0^t \gamma_{\text{land}} v_\psi\big(\tau,\, z^{\hat{y}_c}(\tau),\, \hat{y}_c\big)\, d\tau, \quad t \in [0, 1].$$

$z_{\text{land}} \leftarrow z^{\hat{y}_c}(1)$

**Output:**   Transported point $z_{\text{land}}$

---

**Algorithm**   We define the combination of a lifting transport and a landing transport as a LEAP (see Algorithm 1). Specifically, a *Blending Leap* uses step sizes $\gamma_{\text{b}} = \gamma_{\text{b,lift}} = \gamma_{\text{b,land}} < 1$, while an *Injection Leap* sets $\gamma_{\text{i,lift}} < \gamma_{\text{i,land}}$. Based on these, we introduce our Algorithm 2: LEAPFACTUAL, which combines $N_{\text{b}}$ blending leaps and $N_{\text{i}}$ injection leaps. Generally, a small step size is preferred

**Algorithm 2** LEAPFACTUAL

---

Input: Source point $x$, target label $\hat{y}_c$, classifier $f_\theta$, generative model $g_\phi$, flow matching model $v_\psi$ trained with CE-CFM objective

Hyperparameters: Number of blending leaps $N_\text{b}$, blending step size $\gamma_\text{b}$, number of injection leaps $N_\text{i}$, injection step sizes $\gamma_\text{i, lift} < \gamma_\text{i,land}$

Step 1: Preprocessing
      $z \leftarrow g_\phi^{-1}(x)$             ▷ Acquiring $z$ via VAE encoder or GAN inversion
      $y_\text{c} \leftarrow f_\theta(g_\phi(z))$             ▷ Determining the current label

Step 2: Information Blending             ▷ Generating CE
      **for** j = 0 to $N_\text{b} - 1$ **do**
           $z \leftarrow$ LEAP$(v_\psi, z, y_c, \hat{y}_c, \gamma_\text{b}, \gamma_\text{b})$     ▷ Blending the source and target classes information
           $y_c \leftarrow f_\theta(g_\phi(z))$
      **end for**

Step 3: (Optional) Information Injection             ▷ Generating Reliable CE
      **for** j = 0 to $N_\text{i} - 1$ **do**
           $z \leftarrow$ LEAP$(v_\psi, z, y_c, \hat{y}_c, \gamma_\text{i,lift}, \gamma_\text{i,land})$     ▷ Injecting the target class information
           $y_c \leftarrow f_\theta(g_\phi(z))$
      **end for**

Step 4: Postprocessing
      $x_\text{CE} = g_\phi(z)$

Output: Transported point $x_\text{CE}$

---

for higher precision, although this may require a larger number of leaps and consequently a higher inference time. The number of steps depends on the dataset and specific target classes. However, when blending only, the algorithm will automatically stop as soon as the target class is reached due to information replacement. Generally, starting with a small step size and a large number of steps is suggested.

## 4  Experiments

In the following sections, we demonstrate the performance of LEAPFACTUAL across diverse datasets and experiment setups. We differentiate between LEAPFACTUAL and LEAPFACTUAL_R, the latter one additionally includes information injection. We refer to CEs generated with LEAPFACTUAL_R as reliable CEs. In Section 4.1, we compare our method with state-of-the-art methods and demonstrate the interpretability of our CEs using the Morpho-MNIST dataset [48]. In Section 4.2, we show that reliable CEs enhance classification performance using the Galaxy 10 DECaLS dataset [49, 50]. Finally, in Section 4.3, we use the FFHQ[51] dataset and StyleGAN3[52] to illustrate scalability and applicability to non-differentiable methods. We evaluate performance based on correctness, Area Under the ROC Curve (AUC) and Accuracy (ACC) using the desired target label as ground truth, and similarity, measures depend on the individual experiments. More experimental details in Appendix B.

### 4.1  Quantitative Assessment

In this section, we compare the performance of our method against two competitors: the Opt-based and CGM-based methods. Morpho-MNIST [48] provides a modified version of MNIST specifically designed to benchmark representation learning. We evaluate similarity as the Absolute Relative Error between morphological properties of counterfactual and reconstructed samples (see Appendix B.2.1). To ensure a fair comparison, we use the same VAE model for both the Opt-based and proposed methods, and apply only minimal modifications to the CGM-based method (refer to Appendix B). We use a 4-layer MLP as the flow matching model and explain a simple MLP classifier, with test accuracy 96.58%. Implementation details are provided in Appendix B.

As shown in Table 2, the Opt-based method yields the poorest performance across both correctness and similarity metrics. This is primarily due to gradient vanishing, which causes counterfactual trajectories to halt before reaching the target decision region. This issue is clearly visible in Figure 5, where many counterfactuals fail to resemble typical samples from the target class. The CGM-based method performs significantly better in terms of correctness but introduces greater distortion compared to the proposed approach. These observations are consistent with the discussion in Section 3.1.

Table 2: Results on Morpho-MNIST for 5 runs across different methods (columns) and evaluation metrics (rows). First rows report Accuracy (ACC) and Area Under the ROC Curve (AUC), while the remaining rows show mean and standard error of Absolute Relative Error (D) in morphological properties. The 1st, 2nd, and 3rd best performances are indicated by **bold**, underline, and *italic*.

| Metric | Opt-based | CGM-based | LeapFactual | LeapFactual_R |
|---|---|---|---|---|
| ACC | 0.8277±0.0071 | *0.9422±0.0045* | 0.9868±0.0026 | **0.9906±0.0016** |
| AUC | 0.8814±0.0061 | *0.9979±0.0002* | 0.9990±0.0002 | **0.9999±0.0000** |
| D(Area) | *0.2475±0.0040* | 0.2556±0.0026 | **0.1665±0.0035** | 0.2295±0.0056 |
| D(Length) | 0.2472±0.0039 | 0.2999±0.0017 | **0.2128±0.0039** | *0.2732±0.0029* |
| D(Slant) | 4.3667±0.2430 | *3.4613±0.1783* | **2.5026±0.2554** | 3.3144±0.3352 |
| D(Thickness) | 0.1724±0.0031 | 0.0859±0.0014 | **0.0807±0.0011** | *0.0895±0.0020* |
| D(Width) | *0.2912±0.0033* | 0.3069±0.0012 | **0.2061±0.0039** | 0.2784±0.0039 |
| D(Height) | 0.0620±0.0005 | 0.0288±0.0003 | **0.0265±0.0007** | *0.0303±0.0007* |

In comparison, LEAPFACTUAL demonstrates superior performance in both correctness and similarity. It generates correctly classified counterfactual samples while minimally modifying the original inputs. The information injection in LEAPFACTUAL_R further improves correctness without compromising similarity. As shown in Figure 5, our methods produce realistic counterfactual samples that modify only the most important class-related features, preserving digit style. This selective modification is particularly beneficial for distinguishing similar classes (e.g., 9 vs. 7 or 5 vs. 8), thereby enhancing interpretability. More qualitative comparisons in Appendix B.

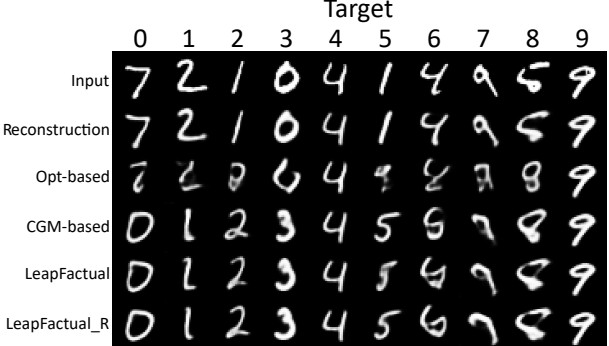

Figure 5: Qualitative comparison of counterfactual samples across different methods (rows) and target labels (columns). The first two rows are randomly sampled target labels and inputs, followed by the reconstructed images.

## 4.2 Model Improvement

We show the advantages of reliable CEs in model training on the Galaxy10 DECaLS dataset [49, 50], a 10-class galaxy morphology classification task. In the experiment, we fix $N_b = 250$, $\gamma_b = 0.1$, $\gamma_{i,\text{lift}} = 1$, and $\gamma_{i,\text{land}} = 1.025$ based on ablation results in Appendix B.

We train a *weak* classifier on 20% of the dataset and use a second VGG model architecture trained on 100% as baseline. We generate standard and reliable CEs (depicted in Figure 6) regarding all classes other than the original for each image in the training subset, resulting in two auxiliary datasets assuming CE target labels as ground truth.

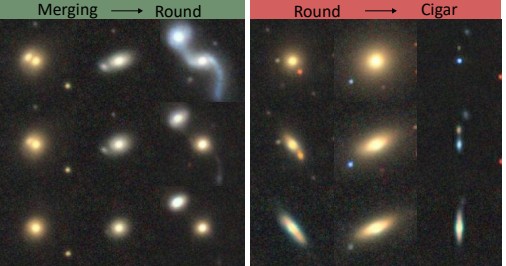

Figure 6: Top to bottom: Input images, CEs, and reliable CEs. **(Left)** *Merging* to *Round* galaxy. **(Right)** *Round* to *Cigar* galaxy.

We then blend varying fractions of these auxiliary datasets with the original training subset, used to train the *weak* classifier (20%), to evaluate their impact on the models classification performance.

Table 3: Model performance across 5 runs with different fractions (second row) of standard and reliable CEs added to training set. Baselines trained with 20% and 100% of training set.

| Metric | Baseline | | CE | | | Reliable CE | | |
|---|---|---|---|---|---|---|---|---|
| | 20% | 100% | 10% | 50% | 100% | 10% | 50% | 100% |
| ACC↑ | 0.8107 ± 0.0041 | 0.8530 ± 0.0010 | 0.7975 ± 0.0016 | 0.7968 ± 0.0017 | 0.7967 ± 0.0022 | 0.8163 ± 0.0020 | 0.8197 ± 0.0016 | 0.8237 ± 0.0021 |
| AUC↑ | 0.9770 ± 0.0005 | 0.9813 ± 0.0003 | 0.9749 ± 0.0002 | 0.9744 ± 0.0001 | 0.9738 ± 0.0003 | 0.9776 ± 0.0002 | 0.9788 ± 0.0004 | 0.9793 ± 0.0005 |

When standard CEs are blended into the training data, model performance declines as the proportion of CEs increases, shown in Table 3. In contrast, using reliable CEs leads to improvements in both accuracy and AUC with increasing fraction. This is because the standard CEs are on the learned decision boundary of the weak classifier, while reliable CEs align closely with the true decision boundary (see Appendix B.1). Thus, they can serve as an augmentation strategy for small datasets or imbalanced classes, thereby addressing fairness and reducing the underrepresentation of minorities. Further, reliable CEs can improve model validation by keeping class-unrelated features similar to the original input, improving expressiveness and helping to identify shortcut learning [53, 54]. We will further investigate these promising results in future work.

## 4.3 Generalization

In this experiment, we demonstrate that the proposed method is highly scalable and applicable to non-differentiable classifiers, such as human annotators. To simulate this scenario, we employ a pretrained CLIP model [55] as a proxy for human judgment. The model is used to classify images as either *Smiling Face* or *Face*. The images are generated by a StyleGAN3 model [52] pretrained on the FFHQ dataset [51] ($1024^2$ pixels). In this setting, existing Opt-based and CGM-based methods are infeasible due to their reliance on differentiability or retraining.

Table 4: Quantitative results for FFHQ using 1,024 samples. Last row reports results for randomly paired images.

| $N_b$ | ACC↑ | AUC↑ | SSIM↑ | PSNR↑ | LPIPS↓ |
|---|---|---|---|---|---|
| 5 | 0.706 | 0.706 | 0.564 | 18.036 | 0.149 |
| 10 | 0.957 | 0.957 | 0.538 | 17.126 | 0.171 |
| 15 | 0.982 | 0.982 | 0.531 | 16.946 | 0.176 |
| 20 | 0.993 | 0.993 | 0.525 | 16.833 | 0.180 |
| - | - | - | 0.070 | 8.848 | 0.555 |

For the flow matching model, we train a 1D U-Net [56] for 120 epochs. A total of 20K images are randomly sampled from StyleGAN3 and projected into the *w*-space [51], which serves as input. The corresponding predicted labels from the CLIP model are used as conditional inputs.

In Figure 7, rows 1 and 2 illustrate image transformations from *Face* to *Smiling Face*, and vice versa, using our method. Note, even subtle differences in facial features lead to changes in classification outcomes. Comparing rows 2 and 3 in Figure 7, we observe that the target feature –facial expression– is further accentuated, providing a clear visual explanation. Table 4 presents accuracy and similarity metrics for various values of $N_b$.

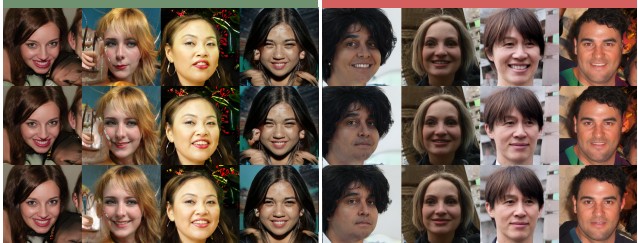

Figure 7: Top to bottom: randomly sampled input images, CEs, and reliable CEs. (**Left**) *Face* to *Smiling Face*. (**Right**) *Smiling Face* to *Face*.

## 5 Conclusion

Our work addresses key limitations in existing counterfactual explanation methods by introducing LEAPFACTUAL, a novel algorithm comprising the CE-CFM training objective and explanation procedure. Our analysis shows that LEAPFACTUAL offers a flexible mechanism for generating counterfactual explanations, capable of producing not only high-quality but also reliable results, even in the presence of discrepancies between true and learned decision boundaries. While our current focus is on visual data, the method is broadly applicable to other data modalities as well.

**Limitations & Future Work** A limitation arises from the dimensionality of the latent space. In high-dimensional settings, such as those involving diffusion models or normalizing flows, the flow matching model cost increases significantly, leading to reduced training efficiency. Future work includes improving LEAPFACTUAL by replacing I-CFM with more efficient approaches like OT-CFM [42]. However, this improvement is not trivial, since the application of OT-CFM requires modifications to the transport map on the classifiers' predictions. Furthermore, we see potential for future work exploring latent space entanglement and challenges associated with imbalanced datasets.

**Acknowledgement** This work was partially funded by project W2/W3-108 Initiative and Networking Fund of the Helmholtz Association. We are grateful to Andrew Alexander Draganov for his constructive discussions and thorough proofreading, which substantially enhanced the clarity and presentation of this work.

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

# A Proofs

## A.1 Conditional Independence

*Proof.* We define the joint latent variable representation as $h = (z_0, z_1 \mid c, x)$, where $x$ denotes the input image and $c = f_\theta(x)$ is the corresponding classifier output. Here, $z_1$ represents the latent encoding of $x$, and $z_0$ is sampled from a Gaussian prior. The distribution over $h$ can be expressed as:

$$\begin{aligned}
q(h) &= q(z_0, z_1 \mid c, x) \\
&= \frac{\int q(z_0, z_1, c, x, r)\, dr}{q(c, x)} \\
&= \int q(r \mid x, c)\, q(z_0 \mid r)\, q(z_1 \mid r, c)\, dr \\
&= q(z_0)\, q(z_1 \mid c)
\end{aligned}$$

The transition from the second to the third line relies on the conditional dependencies illustrated in Figure 3. From the third to the fourth line, we use the fact that the auxiliary variable $r$ is fully determined by $x$ and $c$. Furthermore, since $z_1$ is an encoded representation of $x$, we assume that all relevant information in $x$ is captured by $z_1$. Consequently, the conditioning on $x$ can be omitted in the final expression. For the same reason, we remove the $x$-conditioning in the representation and define $h = (z_0, z_1 \mid c)$.

$\square$

## A.2 Mutual Information

*Proof.* We adopt the information bottleneck framework, which aims to find a representation $Z$ that balances compression of the input $X$ with retention of information relevant to the target label $Y$. The objective is:

$$\min \quad I(X; Z) - \beta I(Z; Y),$$

where $I(X; Z)$ quantifies compression and $I(Z; Y)$ quantifies predictive power, with $\beta$ as the trade-off coefficient.

In our case, $Z_1$ is lifted to $Z_0$ conditioned on the class $C$. The goal of lifting is to minimize $I(Z_0 \mid C; Z_1)$, effectively compressing $Z_1$ into $Z_0$ while conditioning on class-related information. Conversely, the landing transport reconstructs $Z_1$ from $Z_0$ conditioned on $C$, aiming to maximize $I(Z_0; Z_1 \mid C)$. Since the transport is invertible, we assume $\beta = 1$.

The information loss is expressed as:

$$\begin{aligned}
&\min_{\psi} \quad I(Z_0 \mid C; Z_1) - I(Z_0; Z_1 \mid C) \\
&= \min_{\psi} [I(Z_0, Z_1, C) - I(Z_1, C)] - [I(Z_0, Z_1, C) - I(Z_0, C)] \\
&= \min_{\psi} I(Z_0, C) - I(Z_1, C) \\
&= - C.
\end{aligned}$$

where $\psi$ parameterizes the vector field used in flow matching. In the last two steps, we notice from Figure 3 that the minimum mutual information between $Z_0$ and $C$ is zero because $Z_0$ only depends on $R$ in optimal setting; the maximum mutual information between $Z_1$ and $C$ is $C$ because $Z_1$ contains both $C$ and $R$. This derivation confirms that the information removed during lifting corresponds precisely to $C$, the class-related information. $\square$

# B Experiments

The following section provides more details regarding experiments including the implementation (Section B.1) and evaluation metrics (Section B.2).

## B.1 Implementation Details

This section provides more details regarding compute resources, datasets and hyperparameters.

### B.1.1 Technical Details

We performed our experiments on a single node of a GPU server, which includes one NVIDIA A100 with 80GB of VRAM, and an AMD EPYC 7742 with 1TB RAM shared with the other nodes of the server.

### B.1.2 Quantitative Assessment

**Morpho-MNIST** Morpho-MNIST [48] is modified version of MNIST. The dataset is designed as a benchmark dataset aimed at quantitatively assessing representation learning.

**Generative Models** The VAE model consists of an encoder and a decoder. The encoder architecture is defined as follows: Conv(3, 8, 3, 2, 1) $\to$ BN $\to$ LeakyReLU(0.2) $\to$ Conv(8, 16, 3, 2, 1) $\to$ BN $\to$ LeakyReLU(0.2) $\to$ Conv(16, 32, 3, 2, 1) $\to$ BN $\to$ LeakyReLU(0.2) $\to$ Conv(32, 32, 3, 2, 1) $\to$ BN $\to$ LeakyReLU(0.2) $\to$ Linear(128, 64), where the final linear layer produces a 64-dimensional vector, which is split into the mean vector $\mu$ and the standard deviation vector $\sigma$. The decoder mirrors the encoder architecture and includes an additional $1 \times 1$ convolutional layer with a Sigmoid activation at the end to reconstruct the input data.

The model is trained for 100 epochs using the Adam optimizer with a learning rate of $5 \times 10^{-3}$ and a batch size of 256. The loss function combines the mean squared error (MSE) reconstruction loss and the Kullback–Leibler divergence (KLD), with a weighting ratio of 4,000 between the MSE and the KLD terms.

**Opt-based Method** For each batch of input, we optimize Equation (1) using the Adam optimizer with a learning rate of 0.2 for 1,000 epochs. The hyperparameter $\lambda$ is set to 0.0006 to mitigate gradient *vanishing*.

We also experimented with ReLU activation functions in the VAE architecture. However, the results show that the counterfactual explanations (CEs) remain unchanged due to gradient *decay*. Consequently, we retain the LeakyReLU activation with a negative slope of 0.2 throughout the experiment.

**CGM-based Method** To incorporate conditional information, we extend the input dimension of the linear layer at the end of the encoder and the linear layer at the beginning of the decoder from 128 to 138. This allows the model to accept a 10-dimensional one-hot encoded class label from the Morpho-MNIST dataset as the condition input. The training procedure remains the same as that of the standard VAE model.

**LeapFactual** The architecture of the flow network is defined as follows: Linear(32 + 1 + 10, 64) $\to$ SiLU $\to$ Linear(64, 64) $\to$ SiLU $\to$ Linear(64, 64) $\to$ SiLU $\to$ Linear(64, 32), where the input dimensions 32, 1, and 10 correspond to the latent vector from the VAE, a time conditioning variable, and a one-hot encoded class label, respectively. The flow matching noise parameter $\sigma$ is set to 0. The model is trained using the Adam optimizer with a learning rate of 0.005 and a batch size of 256 for 30 epochs, taking approximately 80 seconds to complete.

For this experiment, the hyperparameters of LEAPFACTUAL are set as follows: $\gamma_b = 0.1$ and $N_b = 15$ for blending, and $\gamma_{i, lift} = 0$, $\gamma_{i, land} = 0.1$, and $N_i = 5$ for injection.

Additional qualitative results from LEAPFACTUAL are provided in Figure 8.

Figure 8: Morpho-MNIST: different input (rows) and target labels (columns). In each column, the input, CEs, and reliable CEs are shown from left to right.

Table 5: Galaxy10 ablation experiment of $N_b$ with fixed $\gamma_b = 0.1$

| $N_b$ | ACC↑ | AUC↑ | ACC$_r$↑ | AUC$_r$↑ | SSIM↑ | PSNR↑ | LPIPS↓ |
|---|---|---|---|---|---|---|---|
| 100 | 0.9718 | 0.9855 | 0.4868 | 0.8798 | **0.8579** | **22.8416** | **0.0641** |
| 150 | 0.9772 | 0.9884 | 0.4862 | 0.8829 | 0.8562 | 22.3836 | 0.0655 |
| 200 | 0.9814 | 0.9896 | 0.4880 | 0.8832 | 0.8551 | 22.1526 | 0.0663 |
| 250 | **0.9832** | **0.9901** | **0.4892** | **0.8836** | 0.8544 | 21.9950 | 0.0668 |
| 300 | 0.9832 | 0.9898 | 0.4874 | 0.8835 | 0.8538 | 21.8846 | 0.0673 |

### B.1.3 Model Improvement

This experiment deals with Model improvement.

**Dataset** The Galaxy10 DECaLS dataset is publicly available: Galaxy10 DECaLS. It includes around 18,000 colored images and deals with a 10-classes galaxy morphology classification task. The dataset is split into training and test sets with a fraction of 90% and 10%. The pre-processing includes random rotation augmentation, cropping to the center $150 \times 150$ pixels, and resizing to $128 \times 128$.

**VAE Model** The architecture of the VAE is adapted from Ditria and Drummond [57], with all ELU activation functions replaced by LeakyReLU. For latent space regularization, we substitute the KL divergence penalty with the Maximum Mean Discrepancy (MMD) [58].

**Flow Model** The architecture of the flow network is defined as follows: Linear(64 + 1 + 10, 256) $\rightarrow$ SiLU $\rightarrow$ Linear(256, 256) $\rightarrow$ SiLU $\rightarrow$ Linear(256, 256) $\rightarrow$ SiLU $\rightarrow$ Linear(256, 256) $\rightarrow$ SiLU $\rightarrow$ Linear(256, 256) $\rightarrow$ SiLU $\rightarrow$ Linear(256, 64), where the input dimensions 64, 1, and 10 correspond to the latent vector from the VAE, the time conditioning variable, and the one-hot encoded class label, respectively. The flow matching noise parameter $\sigma$ is set to 0. The model is trained using the Adam optimizer with a learning rate of $10^{-4}$, a batch size of 256, and for a total of 500 epochs. Training takes approximately 2 hours.

**Ablation Study** We begin with an ablation study to identify the optimal value of $N_b$ for blending, while keeping $\gamma_b = 0.1$ fixed. As shown in Table 5, the best performance is achieved with $N_b = 250$. Next, we fix $N_b = 250$ and $\gamma_b = 0.1$, and proceed to tune the injection hyperparameters.

To evaluate reliable counterfactual CEs, we train two VGG16 classifiers using 20% and 100% of the dataset, referred to as the *weak* and *strong* classifiers, respectively. Their test accuracies are approximately 81% and 85%. We generate both standard and reliable CEs from the **weak** classifier and evaluate them using accuracy (ACC) and area under the curve (AUC) metrics, as assessed by the **strong** classifier. These are denoted as ACCr and AUCr, respectively. These two metrics are used to select the best injection hyperparameters for generating reliable CEs.

To tune the injection parameters, we first fix the difference $\gamma_{i,land} - \gamma_{i,lift}$ and vary $\gamma_{i,lift}$, followed by optimizing $\gamma_{i,land}$. As shown in Figure 9, the best performance is obtained with $\gamma_{i,lift} = 1$ and $\gamma_{i,land} = 1.025$.

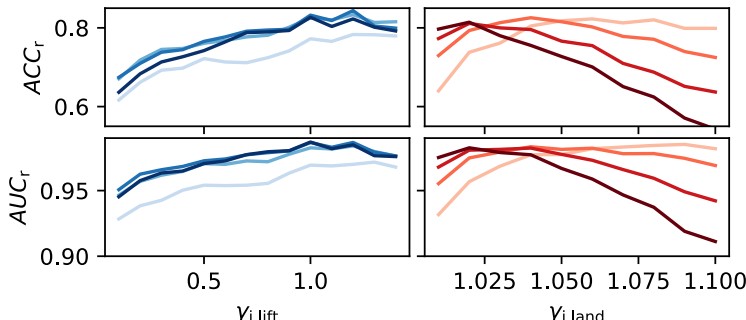

Figure 9: Hyperparameter ablation (**Left**) Ablation of $\gamma_{i,lift}$ with $\gamma_{i,land} - \gamma_{i,lift} = 0.025$. (**Right**) Ablation of $\gamma_{i,land}$ with $\gamma_{i,lift} = 1$. Line color intensity increases with $N_i$ in steps of 10, starting at 10.

Finally, we observe the effect of $N_i$. In Table 6, we can find that reliable metrics improve with injection steps. Notably, $ACC_r$ rises from 47% to 82% without degrading the weak classifier's accuracy. This confirms that the standard CE is lying around the weak classifier's decision boundary, while reliable CE is more aligned with the true decision boundary.

More qualitative results are shown in Figure 10.

Table 6: Quantitative results for Galaxy10 reliable CEs. Uncertainties calculated across 10 runs.

| Metric | $N_i = 0$ | $N_i = 5$ | $N_i = 15$ | $N_i = 25$ | $N_i = 35$ |
|---|---|---|---|---|---|
| ACC↑ | **0.9821 ± 0.0008** | 0.9439 ± 0.0017 | 0.9599 ± 0.0011 | 0.9612 ± 0.0012 | 0.9545 ± 0.0011 |
| AUC↑ | 0.9897 ± 0.0003 | 0.9934 ± 0.0003 | 0.9973 ± 0.0001 | **0.9975 ± 0.0001** | 0.9974 ± 0.0001 |
| $ACC_r$↑ | 0.4738 ± 0.0034 | 0.6620 ± 0.0040 | 0.7899 ± 0.0023 | **0.8171 ± 0.0028** | 0.8135 ± 0.0031 |
| $AUC_r$↑ | 0.8797 ± 0.0012 | 0.9404 ± 0.0009 | 0.9740 ± 0.0006 | 0.9815 ± 0.0005 | **0.9832 ± 0.0005** |
| SSIM↑ | **0.8577 ± 0.0007** | 0.8255 ± 0.0009 | 0.7774 ± 0.0010 | 0.7417 ± 0.0010 | 0.7122 ± 0.0010 |
| PSNR↑ | **22.4959 ± 0.0778** | 21.0064 ± 0.0826 | 19.2938 ± 0.0768 | 18.0882 ± 0.0787 | 17.0410 ± 0.0733 |
| LPIPS↓ | **0.0651 ± 0.0005** | 0.0821 ± 0.0006 | 0.1095 ± 0.0006 | 0.1324 ± 0.0007 | 0.1524 ± 0.0007 |

#### B.1.4 Generalization

This section deals with the experiment setup regarding the generalization experiment.

**CLIP** We use a CLIP model with a Vision Transformer (ViT-B/32) as the image encoder. The data is transferred from [-1, 1] to [0, 1] before being fed into the transformation of the CLIP model. The model is trained to differentiate between face and smiling face. We determine the labels based on the higher score among Smiling face and Face predicted by the CLIP mode.

**StyleGAN3** We use a StyleGAN3 [52] pretrained on the FFHQ dataset (checkpoint name stylegan3-r-ffhq-1024x1024.pkl from here). The noise mode is set to 'const' and the truncation_psi is set to 1. We randomly sample from a Gaussian distribution and project them to w-space and image space via the mapping function and decoder.

**LeapFactual** In this experiment, we use a 1D U-Net [56] as the flow matching model. To train the model, we first sample 20,000 random latent vectors from a Gaussian distribution and map them to the $w$-space as the input. Then we use the classifier output of corresponding images as the condition of the U-Net. We use Adam optimizer to train the model for 120 epochs with a batch size of 32, a learning rate of $2 \times 10^{-4}$, and a weight decay of $1 \times 10^{-5}$. Training is performed on a single NVIDIA A100 GPU and takes approximately 22 hours.

The hyperparameters for the blending parameters are $\gamma_b = 0.8$ and $N_b = 10$. For information injection, we set $\gamma_{i,lift} = 0.8$, $\gamma_{i,land} = 0.83$, and $N_i = 5$. The flow matching $\sigma$ is set to $10^{-4}$.

### B.2 Evaluation Metrics

This section introduces the evaluation metrics we employed for the evaluation of our experiments.

### B.2.1 Morphometrics

[48] proposed the Morpho-MNIST dataset as well as measurable properties known as morphometrics to evaluate differences regarding characteristics like stroke thickness and area, length, width, height and slant of the depicted digits. The assessment of these metrics begins with first binarizing the image, followed by measuring various features about the resulting skeleton including: the length of the skeleton (length), average distance between the center of skeleton and its closest borders (thickness), horizontal shearing angle (slant) and the dimensions of a bounding box (width, height and area).

We report the absolute of the relative error regarding the morphometrics:

$$\text{Absolute Relative Error} = \left| \frac{\text{Absolute Error}}{\text{True Value}} \right|,$$

with the absolute error being the difference between measured and true value. We employ the absolute value, since the angle reported for slant may be negative.

We used the official implementation available online: Morpho-MNIST git. For more details, please refer to [48].

### B.2.2 Metrics

We assess the correctness of the generated CE employing accuracy (ACC) and area under the Receiver Operating Curve (AUROC), using the implementations provided by torchmetrics. For both metrics higher is better. While accuracy provides a quick overview of how often the model provides the correct prediction (correct predicitons / total predictions). Since AUROC denoted the area under the curve between true positive rate and false positive rate, it provides a deeper insight and is favourable especially when the dataset is imbalanced.

To evaluate the similarity between original image and CE, we employ three well-known Perceptual Similarity Metrics working on different levels. Including Structural Similarity Index (SSIM) [59], Learned Perceptual Image Patch Similarity (LPIPS) metric [60], and Peak Signal-to-Noise Ratio (PSNR).

SSIM evaluates similarity based on comparing three concepts: luminance, structure and contrast.

$$\text{SSIM}(x, x_{\text{CE}}) = \frac{\left(2\mu_x \mu_{x_{\text{CE}}} + C_1\right)\left(2\sigma_{xx_{\text{CE}}} + C_2\right)}{\left(\mu_x^2 + \mu_{x_{\text{CE}}}^2 + C_1\right)\left(\sigma_x^2 + \sigma_{x_{\text{CE}}}^2 + C_2\right)} \tag{4}$$

The influence of luminace is included as the means of pixel values denoted as $\mu_x$ and $\mu_{x_{\text{CE}}}$, contrast is included with the standard deviations $\sigma_x$ and $\sigma_{x_{\text{CE}}}$ and finally the covariance $\sigma_{xx_{\text{CE}}}$ is employed to take account for structure.

PSNR, in contrast, is a more simple metric building on the mean-squared error and comparing on a pixel-wises basis.

$$\text{PSNR}(x, x_{\text{CE}}) = 10 * \log_{10}\left(\frac{\max(x)^2}{\text{MSE}(x, x_{\text{CE}})}\right) \tag{5}$$

Unlike PSNR or SSIM, LPIPS evaluates the similarity of two images based on deep features extracted from a neural network, as a feature extractor we set Squeezenet. The similarity is evaluated by comparing layer-wise computed euclidean distances of the normalized images. Which are subsequently being accumulated as a weighted-sum, the weights being learned. The score can then denotes the distance between the two images, therefore a lower LPIPS score resembles a higher similarity.

More details can be found in the original publications. In our experiments, we use the implementations provided by torchmetrics, setting Squeezenet as backbone network for LPIPS while keeping all other parameters at their default values.

## C   Broader Impacts

Our work introduces a method to generate realistic and reliable counterfactuals, aiming at interpretability. By demonstrating how input data has to be modified in order to obtain a different decision and providing actionable changes, we promote transparency and reduce of barriers to the application of Machine Learning models.

Potential positive impacts of our approach may include, supporting non-expert users in understanding decisions, enhancing scientific knowledge discovery and enabling future works regarding fairness. Our method could be extended to generate counterfactual samples for underrepresented minorities in datasets. Additionally, counterfactual explanations can be utilized to investigate features contributing to the decision, thereby facilitating bias detection.

However, there are risks for potential misuse, such as the creation of deep fakes. Our approach allows changing the input data towards specific classes, thus allowing to generate new data. Another concern may be that a simplified representation of the decision-making process could mislead users. While we have no opportunity to mitigate the risk towards deep fakes, we advocate for the use of reliable CEs to minimize potential oversimplifications. These CEs ensure, that the generated counterfactual sample remain in the distribution of the data, avoiding out-of-distribution artifacts that could mislead the user.

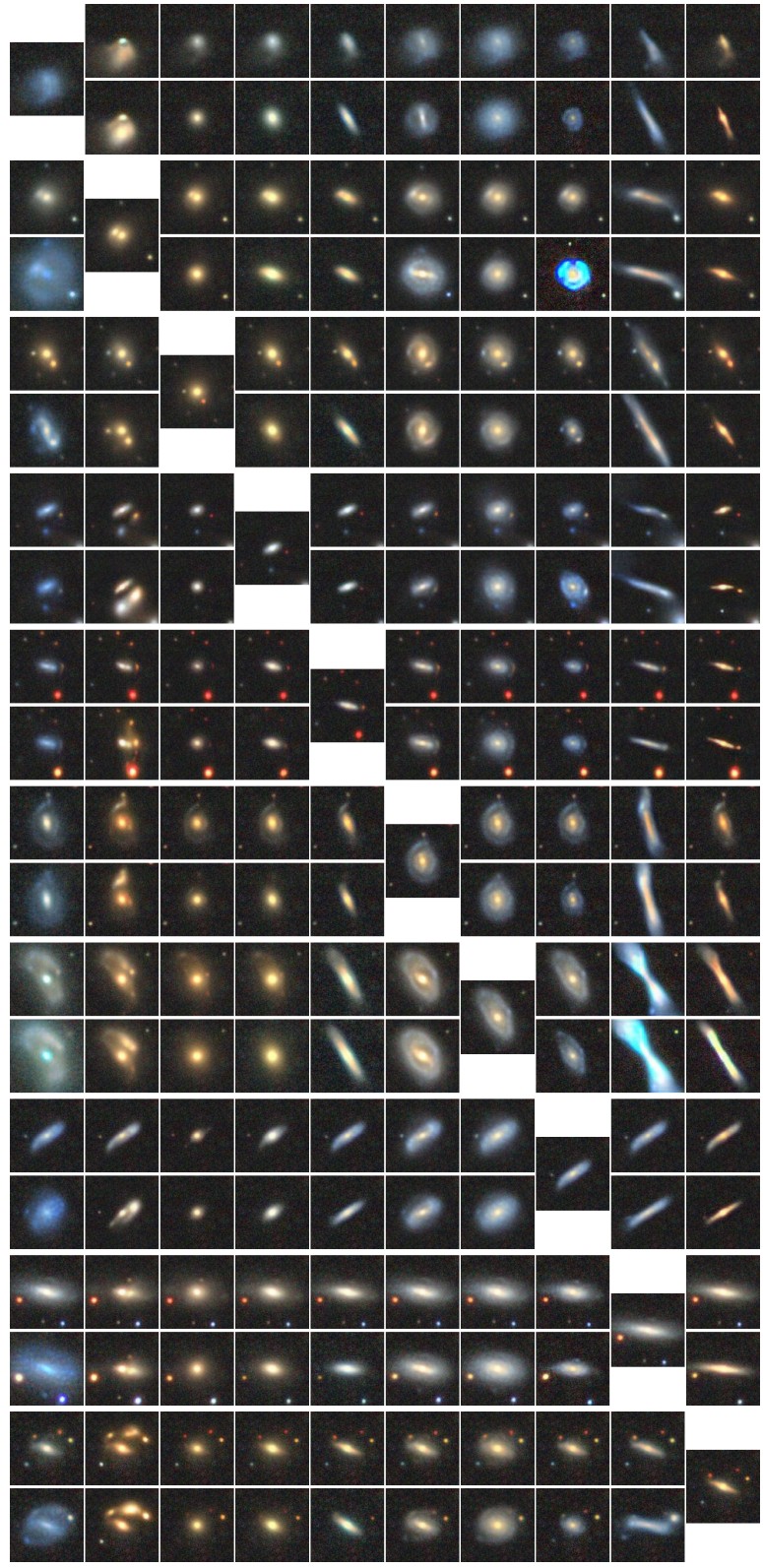

Figure 10: In each group (two rows), the column with a single image is the input, the remaining are CEs (top) and reliable CEs (bottom). From top to bottom, the input image are in the classes of *Distributed*, *Merging*, *Round Smooth*, *In-between Round Smooth*, *Cigar Round Smooth*, *Barred Spiral*, *Unbarred Tight Spiral*, *Unbarred Loss Spiral*, *Edge-on without Bulge*, *Edge-on with Bulge*

