# OpenReview forum: "LeapFactual: Reliable Visual Counterfactual Explanation Using Conditional Flow Matching"
_NeurIPS.cc/2025/Conference — NeurIPS 2025 poster_

### Official Review · Reviewer_8euj · 2025-06-30

**Clarity:** 2
**Significance:** 3
**Originality:** 3
**Rating:** 4
**Confidence:** 3

**Summary:**

LeapFactual is a new method for generating visual counterfactual explanations leveraging conditional flow matching on an input's latent representation. Counterfactual explanations promise to provide more insight into a classifier's decision, since they try to apply (minimal) changes to an input $x$, transforming it to $x_\mathrm{CE}$, such that the classifier $f$ predicts target class $c^\prime = f(x_\mathrm{CE})$ instead of the original class $c = f(x)$. If the intrinsic dimension of the data is $<<$ than its ambient dimension, it is common to use a generative model $g$ to transform $x$ to its lower-dimensional latent representation $z = g^{-1}(x)$ and search an appropriate $z_\mathrm{CE}$ in the latent space such that $x_\mathrm{CE} = g(z_\mathrm{CE})$.
The authors claim that with their flow-based approach to counterfactual generation, they mitigate the drawbacks of the two predominant approaches:
1. Optimization-based approaches (OPT-based) that minimize an objective composed of the classifier loss term for the new _target_ class $c'$ and a distance term between $x_\mathrm{CE}$ and $x$, aiming to steer $x \to x_\mathrm{CE}$ such that the predicted class label flips to the desired class while keeping $x$ and $x_\mathrm{CE}$ similar. While enabling continuous latent space interpolation, these approaches seem to suffer from vanishing gradients near the decision boundary and necessitate differentiable classifiers.
2. Conditional Generative Model-based approaches (CGM-based), which integrate the classifier information directly by training conditioned on classifier output. This offers structured representations of residual and class information but lacks a continuous latent space due to explicit conditioning on classifier outputs.
The authors achieve this by transforming class-specific latent representations $z_{1,c}$ to class-agnostic latent representations $z_0$ using independent conditional flow matching (I-CMF), a process they refer to as "lifting", and infusing class information back by inverting this flow and conditioning on the target class, which they refer to as "landing". To avoid a situation where $x_\text{CE}$ is too close to the classifier boundary (which might not conform with the true boundary), the authors describe to _inject_ additional class information after solely _blending_ it before.
The authors demonstrate the efficacy of their method on the Morpho-MNIST dataset, which provides a quantitative approach to measuring distortions of class-related and residual information, showcase its usefulness in dataset augmentation using the Galaxy10DECaLS dataset, and its aptitude to setting where classifiers are non-differentiable, reporting positive results.

**Questions:**

1. As alluded to in the "Strengths & Weaknesses" section, I don't quite understand the trade-offs of choosing certain hyperparameters. For instance, in Figure 1, the MNIST example seems to suggest a single leap is sufficient, while in Figure 4, $N_b$ and $N_i$ are higher. For instance, is the choice of $N_b$ and $\gamma_b$ truly independent, or is there an equivalence for certain combinations of parameters?
2. I wonder how this method extends to other data, for instance, textual data, when generating counterfactuals. Are the adaptations as trivial as suggested by the authors?

**Ethical Concerns:**

["NO or VERY MINOR ethics concerns only"]

**Final Justification:**

The rebuttal satisfactorily addressed most of my concerns. The clarification on Figure 4, planned improvements to readability, and corrections to wording improve clarity. The explanation of hyperparameter trade-offs is helpful, though I still see value in adding more concrete heuristics or equivalence insights. The additional tabular dataset experiment meaningfully strengthens the claim of generalizability beyond visual data, though I would like clarification on whether one-hot categorical constraints are preserved in generated counterfactuals. Given that these remaining points are relatively minor compared to the strengths demonstrated, I maintain a positive overall assessment.

**Limitations:**

Besides the applicability to other data as claimed by the authors, the limitations raised by the authors (primarily the high computational effort due to the CFM approach) seem comprehensive.

**Quality:**

2

**Strengths And Weaknesses:**

Strengths, in no particular order:
1. The authors were able to empirically demonstrate that their method produces counterfactuals that are reliably recognized by the classifier as belonging to the target class while reducing distortions of residual information. This is particularly clearly demonstrated in section 4.1 (Morpho-MNIST task).
2. Their method allows for considerable customization via the various degrees of information replacement, blending, and injection, although some more clarity with regard to the impact of the corresponding hyper-parameters would be appreciated.
3. Generally, the figures and diagrams add some clarity, although Figure 4 could be improved substantially.

---
Weaknesses, in no particular order:
1. Figure 4 and the accompanying text is hard to read:
	1. In the text, the authors state "that multiple points from different classes in $Z_0$ are mapped to the same point in $Z_1$, which confirms Theorem 1". While this observation complies with the theorem, it does not _confirm_ it (this observation would also be true for the trivial mapping that maps all $z_0$ to the origin, which is not a helpful mapping in this case).
	2. The described phenomena in 4 (a)-(d) are hardly visible due to the small size of the subplots. I recommend adjusting the margins between subplots and removing or reducing the size of the tick labels.
2. The method is parameterized by multiple hyper-parameters, $N_b$, $\gamma_b$, $N_i$, and $\gamma_{i,\mathrm{lift}} < \gamma_{i,\mathrm{land}}$. The authors provide some insights into the impact of these hyper-parameters in B.1.3 for the "Model Improvement" task presented in section 4.2, however, it would be interesting to see more ablations for other datasets, including the Morpho-MNIST dataset (and potentially a non-visual dataset, see the following point), or heuristics on how to choose there parameters based on the problem at hand.
3. The authors state that "while our method focuses on visual data, it can be generalized to other data types". However, the authors only provide empirical evidence of the efficacy of their method on visual data (besides the simple toy problem discussed in 3.2.2). I would appreciate an experiment that puts this statement to the test on some non-visual dataset, or remove it from the paper.
4. The clarity of the text could be improved in multiple sections:
	1. In section 2.1, the authors choose to start multiple sentences with bare citation numbers (e.g., "[14] introduced REVISE ..." or "[17] use a GAN ..."), which does not aid clarity.
	2. in line 220, I would favor "subtracting C from X to get R" instead of "deducted"-

---

> ### Author Rebuttal · Authors · 2025-07-29
>
> Dear Reviewer,
>
> We thank you for your time and thorough review. We are very happy to see that most of the issues seem to be solved straightforwardly. To address your concerns, we conducted an extra experiment on the \textbf{tabular adult dataset} and appended it at the end. In the following sections, we address weaknesses and questions individually.
>
> ## Weakness:
> **W1 (Figure 4 - confirm Theorem 1, subplots too small):** Thank you for acknowledging that the included Figures and diagrams added to clarity. We see the issue with the size of the subplots in Figure 4 and will make use of the additional content page in the camera-ready version to extend the Figure. We appreciate the recommendation of how we can improve it.  In the sentence "...which confirms Theorem 1", we recognize that confirm is the wrong choice of word for the description, we will change it to “...which complies with Theorem 1.”
>
> **W2 (hyperparameter ablations for other datasets -> heuristics for choice):** Please see Q1 below.
>
> **W3 (generalization to other datatypes, either non-visual dataset or remove from paper):** Please see Q2 below.
>
> **W4 (Section 2.1 bare citation start + phrasing in line 220):** Thank you for pointing this out. We agree with you and see that the paper's clarity and readability will benefit from rephrasing the sentences in Section 2.1 as well as changing the wording to the suggested one in line 220.
>
> ## Questions:
> **Q1 (hyperparameter choosing; Number of steps in Figure 1 vs. Figure 4):** We provide some hyperparameter ablations in the appendix (Table 5 & 6, and Figure 9, also Table 4 in the main text). The approach takes two types of hyperparameters: step size and number of steps. Generally, a small step size is preferred for higher precision, although this may require a larger number of steps and consequently a higher inference time. The number of steps depends on the dataset and specific classes you generate the counterfactuals for; therefore, we can not provide the optimal parameters for every scenario. However, when blending only, the algorithm will automatically stop as soon as the target class is reached due to information replacement. Generally, we suggest starting with a small step size and a large number of steps. For future works, we see possible improvement, including adaptive step size. Regarding the figures: Figure 1 is a conceptual figure; in reality, the latent space is high-dimensional and more complex. We will improve the caption of the Figure to avoid misunderstanding.
>
> **Q2 (generalization data types):**  Thank you for the suggestion. We have done an extra experiment on the tabular adult dataset. Please see below for details.
>
> ## Extra experiment
> We conducted an additional experiment using the Adult dataset [1]. The dataset was split into training and test sets using an 80/20 ratio.
>
> To establish baseline classifiers, we first trained a **weak classifier** by introducing label noise—flipping the labels of a fraction of the training data. This model achieved a test accuracy of $0.7093$ and an AUC of $0.6874$. Subsequently, we trained a **strong classifier** on clean data without label noise, achieving a test accuracy of $0.8546$ and an AUC of $0.9039$.
>
> Next, we trained a TVAE [2] as our tabular data generative model using the full dataset. We then trained a flow matching model conditioned on the predictions of the weak classifier to generate counterfactual examples (CE) and reliable counterfactual (we changed the name from robust to reliable according to reviewers' suggestions)) examples, where the target labels corresponded to the flipped predictions. Both the weak and strong classifiers were used to evaluate the generated samples in terms of accuracy and AUC.
>
> For the similarity metric, categorical variables were converted into one-hot representations, while numerical features were normalized. The mean squared error (MSE) was then computed to quantify similarity between the generated and original samples.
> The results of the experiment are summarized in the table below.
>
> From the results, we observe the following:
> - The accuracy of the **weak classifier** improves progressively as the number of blending steps increases, but drops sharply upon performing information injection.
> - The accuracy of the **strong classifier**, in contrast, is initially lower than that of the weak classifier during the blending steps. However, it surges from approximately **73**% to **93**% after only 10 injection steps and ultimately reaches **97**%.
> - The **MSE** increases consistently as the number of blending and injection steps grows.
>
> These findings demonstrate that:
> - The proposed method is effective on tabular data, provided the generative model is capable of capturing the underlying causal relationships.
> - The method is robust to imbalanced datasets (the Adult dataset contains 80% low-income and 20% high-income labels) and can handle noisy data (e.g., human annotation errors), as evidenced by the performance of the weak classifier trained with noisy labels.
> - The reliable (prev. robust) counterfactual examples effectively resolve the misalignment between the learned and true decision boundaries, under the assumption that the strong classifier better approximates the true decision boundary than the weak classifier.
>
>
> | Metric        | N_b=30         | N_b=60         | N_b=90         | N_i=10         | N_i=20         | N_i=30         |
> |---------------|-------------------------|-------------------------|-------------------------|-------------------------|-------------------------|-------------------------|
> | **ACC↑**     | 0.7607 ± 0.0027         | 0.8355 ± 0.0026         | **0.8503 ± 0.0026**     | 0.6984 ± 0.0016         | 0.6686 ± 0.0020         | 0.6389 ± 0.0013         |
> | **AUC↑**     | 0.8522 ± 0.0014         | 0.8989 ± 0.0017         | **0.9082 ± 0.0016**     | 0.8070 ± 0.0011         | 0.7905 ± 0.0015         | 0.7713 ± 0.0011         |
> | **ACC_r↑** | 0.7232 ± 0.0017         | 0.7258 ± 0.0017         | 0.7276 ± 0.0019         | 0.9328 ± 0.0016         | 0.9655 ± 0.0009         | **0.9692 ± 0.0009**     |
> | **AUC_r↑** | 0.7841 ± 0.0019         | 0.7860 ± 0.0018         | 0.7874 ± 0.0021         | 0.9466 ± 0.0019         | 0.9747 ± 0.0011         | **0.9783 ± 0.0011**     |
> | **MSE↓**     | **0.0537 ± 0.0001**     | 0.0544 ± 0.0001         | 0.0546 ± 0.0001         | 0.0688 ± 0.0002         | 0.0717 ± 0.0002         | 0.0731 ± 0.0002         |
>
> *Metrics with mean ± standard error for 5 runs. Bold values represent the best-performing configuration for each metric. The first three columns (N_b) indicate performance changes across blending steps, while the last three columns (N_i) indicate performance changes during injection steps, which continue from the last blending step. The step sizes are $\gamma_b = 0.1$, $\gamma_{i,lift}=1$, and $\gamma_{i,land}=1.1$. ACC and AUC are evaluated using the weak classifier, whereas ACC_r and AUC_r are evaluated using the strong classifier.*
>
> [1] Becker, B. & Kohavi, R. (1996). Adult [Dataset]. UCI Machine Learning Repository.
>
> [2] Lei Xu, Maria Skoularidou, Alfredo Cuesta-Infante, Kalyan Veeramachaneni. Modeling Tabular data using Conditional GAN. NeurIPS, 2019.

---

> > ### Comment · Reviewer_8euj · 2025-08-05
> >
> > Thank you for your detailed rebuttal and for conducting the additional ablation on the Adult dataset.
> >
> > While your response to the hyperparameter selection question addresses my main confusion, I still think the paper could benefit from a more explicit discussion (e.g., in the appendix) of whether certain hyperparameter settings tend to be equivalent in effect (e.g., step size vs. number of steps combinations) and any heuristics you found useful beyond the starting recommendation. Regarding the new tabular experiment, the improvement in strong classifier performance after information injection is clear, and the results support your robustness claims.
> >
> > Overall, most of my original concerns have been addressed, and the additional experiment meaningfully strengthens the work. I would maintain my positive evaluation while noting the above point for clarification in the camera-ready version.

---

> > > ### Author Response · Authors · 2025-08-06
> > >
> > > Dear Reviewer,
> > >
> > > Thank you for your reply and acknowledging our effort! We will happily include more context regarding the hyperparameters in the camera version.

---

### Official Review · Reviewer_XFr6 · 2025-07-02

**Clarity:** 3
**Significance:** 3
**Originality:** 3
**Rating:** 5
**Confidence:** 3

**Summary:**

- This paper proposes a method for generating realistic counterfactual examples. Traditional counterfactual explanation methods, including optimization-based approaches and CGM-based methods using label-conditioned generative models, sometimes fail to produce realistic counterfactual examples.
- The authors introduce a new counterfactual explanation method based on conditional flow matching. This approach trains a map between two types of latent representations: a "flattened representation", where class-related and residual features are entangled, and a "structured representation", where only residuals are entangled. By doing so, it aims to generate counterfactual examples that account for both.
- Numerical experiments demonstrate that the proposed method successfully outputs natural images as counterfactual examples. The authors also show that using CEs generated by their method as new training samples can improve prediction performance.

**Questions:**

- The flow matching model uses a simple 4-layer MLP. This isn't inherently an issue, but could the computation time increase significantly if we use a large neural network for the flow matching model in your proposed method? Please discuss the computation time for training a flow-matching model.
- In the numerical experiments, the proposed method was applied to image data, but I am interested in its applicability to tabular data. In tabular data where features have causal relationships, could the proposed method be used to generate causally aware counterfactual explanations?

**Ethical Concerns:**

["NO or VERY MINOR ethics concerns only"]

**Final Justification:**

The authors have sincerely addressed all of my concerns and questions. While I recommend they revise the manuscript to incorporate their responses, I believe the paper is suitable for publication.

**Limitations:**

yes

**Quality:**

3

**Strengths And Weaknesses:**

- Strengths
	- Research on generating realistic counterfactual examples is highly significant in explainable machine learning. This paper proposes a method to achieve this using a flow-matching model. The approach of using flow matching for counterfactual explanation is original, and I believe the research holds substantial value for the broader machine learning community.
	- The computational experiments validate the proposed method's effectiveness through multi-faceted and appropriate procedures. Specifically, the result that prediction performance improves when we use counterfactual examples generated by the proposed method as training samples is particularly interesting.
	- The paper itself is clearly written and easy to read.
- Weaknesses
	- Many explanations rely on intuition based on specific examples, and the theoretical foundations for how the proposed method overcomes the limitations of existing research seem somewhat weak.
	- While not a fundamental weakness, the term "Robust CE" is used in various contexts within Counterfactual Explanations (CE) and Algorithmic Recourse (AR). For instance, (Dominguez-Olmedo et al., '22) defines "Robust CE" as robustness against uncertainty in input features. Given the diverse definitions of robustness in CE (see (Jiang et al., '24)), a more detailed explanation of what "Robust CE" signifies in this research and what "robustness" implies would be beneficial.


(Dominguez-Olmedo et al., '22) Ricardo Dominguez-Olmedo, Amir H Karimi, and Bernhard Schölkopf. On the Adversarial Robustness of Causal Algorithmic Recourse. Proceedings of the 39th International Conference on Machine Learning, PMLR 162:5324-5342, 2022.

(Jiang et al., '24) Junqi Jiang, Francesco Leofante, Antonio Rago, and Francesca Toni. Robust Counterfactual Explanations in Machine Learning: A Survey, https://doi.org/10.48550/arXiv.2402.01928.

(Dominguez-Olmedo et al., '22) Ricardo Dominguez-Olmedo, Amir H Karimi, and Bernhard Schölkopf. On the Adversarial Robustness of Causal Algorithmic Recourse. Proceedings of the 39th International Conference on Machine Learning, PMLR 162:5324-5342, 2022.
(Jiang et al., '24) Junqi Jiang, Francesco Leofante, Antonio Rago, and Francesca Toni. Robust Counterfactual Explanations in Machine Learning: A Survey, https://doi.org/10.48550/arXiv.2402.01928.

---

> ### Author Rebuttal · Authors · 2025-07-29
>
> Dear Reviewer,
>
> We thank you for the thorough and helpful review. We appreciate the feedback and are happy to address it to improve our paper even more. To address your concerns, we conducted an extra experiment on the **tabular adult dataset** and appended it at the end. In the following, we will address weaknesses and questions individually.
>
> ## Weaknesses:
> **W1 (explanations rely on intuition, theoretical foundations for how the proposed method overcomes the limitations of existing research seem somewhat weak):** Thank you for the feedback. We agree that it would be valuable to clarify this point more explicitly. In our revision, we will add a few sentences to state clearly that we address the limitations by applying flow matching on a flattened (OPT-based) representation and subsequently using it to obtain a structured (CGM-based) representation, which is then leveraged for counterfactual generation. This approach allows us to capitalize on the strengths of both paradigms while mitigating their respective drawbacks. We also provide the theoretical foundations of both paradigms and detail how their advantages are integrated in Section 3.1. Regarding the intuitive explanation, we view this as a key property and strength of counterfactual explanations (CE): CE offers users clear information that helps them intuitively grasp the underlying reasoning behind decisions. If there are additional points we have not yet addressed, we would greatly appreciate further feedback so that we can continue improving the paper.
>
> **W2 ("Robust CE"):** We see that the term “Robust CE” brings up confusion as it denotes other characteristics in different contexts. In our case we want to express that the generated counterfactuals are robust in the sense of alignment to the dataset. After reading the reviews we acknowledge the confusion and rename it to **reliable counterfactuals**, as this expresses the dependability of the method we are aiming at.
>
> ## Questions:
> **Q1 (Computation time for a larger flow-matching model):** We agree that the training and inference time will also depend on the dataset and model applied. We included reference training time in Appendix B.1.2, B.1.3, and B.1.4. The model for the MorphoMNIST, Galaxy, and FFHQ experiments takes about 80 secs, 2 hours, and 22 hours to train on a single A100 GPU, respectively. However, we emphasize that the bottleneck for the FFHQ experiment highly comes from StyleGAN, which is a heavy generative model, taking relatively long to decode the latent vectors. The 1D-UNet model can also be improved to increase efficiency. The inference time is in the scale of seconds or minutes, depending on the problem's difficulty.
>
> **Q2 (Generalization data types):** Thank you for the suggestion. We did an extra experiment on the tabular adult data (see below). The experiment suggests that if the generative model is strong enough to capture the causal relation, we can generate reasonable CEs.
>
> ## Extra experiment
> We conducted an additional experiment using the Adult dataset [1]. The dataset was split into training and test sets using an 80/20 ratio.
>
> To establish baseline classifiers, we first trained a **weak classifier** by introducing label noise—flipping the labels of a fraction of the training data. This model achieved a test accuracy of $0.7093$ and an AUC of $0.6874$. Subsequently, we trained a **strong classifier** on clean data without label noise, achieving a test accuracy of $0.8546$ and an AUC of $0.9039$.
>
> Next, we trained a TVAE [2] as our tabular data generative model using the full dataset. We then trained a flow matching model conditioned on the predictions of the weak classifier to generate counterfactual examples (CE) and reliable counterfactual examples, where the target labels corresponded to the flipped predictions. Both the weak and strong classifiers were used to evaluate the generated samples in terms of accuracy and AUC.
>
> For the similarity metric, categorical variables were converted into one-hot representations, while numerical features were normalized. The mean squared error (MSE) was then computed to quantify similarity between the generated and original samples.
> The results of the experiment are summarized in the table below.
>
> From the results, we observe the following:
> - The accuracy of the **weak classifier** improves progressively as the number of blending steps increases, but drops sharply upon performing information injection.
> - The accuracy of the **strong classifier**, in contrast, is initially lower than that of the weak classifier during the blending steps. However, it surges from approximately **73**% to **93**% after only 10 injection steps and ultimately reaches **97**%.
> - The **MSE** increases consistently as the number of blending and injection steps grows.
>
> These findings demonstrate that:
> - The proposed method is effective on tabular data, provided the generative model is capable of capturing the underlying causal relationships.
> - The method is robust to imbalanced datasets (the Adult dataset contains 80% low-income and 20% high-income labels) and can handle noisy data (e.g., human annotation errors), as evidenced by the performance of the weak classifier trained with noisy labels.
> - The reliable counterfactual examples effectively resolve the misalignment between the learned and true decision boundaries, under the assumption that the strong classifier better approximates the true decision boundary than the weak classifier.
>
>
> | Metric        | N_b=30         | N_b=60         | N_b=90         | N_i=10         | N_i=20         | N_i=30         |
> |---------------|-------------------------|-------------------------|-------------------------|-------------------------|-------------------------|-------------------------|
> | **ACC↑**     | 0.7607 ± 0.0027         | 0.8355 ± 0.0026         | **0.8503 ± 0.0026**     | 0.6984 ± 0.0016         | 0.6686 ± 0.0020         | 0.6389 ± 0.0013         |
> | **AUC↑**     | 0.8522 ± 0.0014         | 0.8989 ± 0.0017         | **0.9082 ± 0.0016**     | 0.8070 ± 0.0011         | 0.7905 ± 0.0015         | 0.7713 ± 0.0011         |
> | **ACC_r↑** | 0.7232 ± 0.0017         | 0.7258 ± 0.0017         | 0.7276 ± 0.0019         | 0.9328 ± 0.0016         | 0.9655 ± 0.0009         | **0.9692 ± 0.0009**     |
> | **AUC_r↑** | 0.7841 ± 0.0019         | 0.7860 ± 0.0018         | 0.7874 ± 0.0021         | 0.9466 ± 0.0019         | 0.9747 ± 0.0011         | **0.9783 ± 0.0011**     |
> | **MSE↓**     | **0.0537 ± 0.0001**     | 0.0544 ± 0.0001         | 0.0546 ± 0.0001         | 0.0688 ± 0.0002         | 0.0717 ± 0.0002         | 0.0731 ± 0.0002         |
>
> *Metrics with mean ± standard error for 5 runs. Bold values represent the best-performing configuration for each metric. The first three columns (N_b) indicate performance changes across blending steps, while the last three columns (N_i) indicate performance changes during injection steps, which continue from the last blending step. The step sizes are $\gamma_b = 0.1$, $\gamma_{i,lift}=1$, and $\gamma_{i,land}=1.1$. ACC and AUC are evaluated using the weak classifier, whereas ACC_r and AUC_r are evaluated using the strong classifier.*
>
> [1] Becker, B. & Kohavi, R. (1996). Adult [Dataset]. UCI Machine Learning Repository.
>
> [2] Lei Xu, Maria Skoularidou, Alfredo Cuesta-Infante, Kalyan Veeramachaneni. Modeling Tabular data using Conditional GAN. NeurIPS, 2019.

---

> > ### Comment · Reviewer_XFr6 · 2025-08-05
> >
> > Thank you for your reply.  Since the authors' responses have addressed my concerns, I will maintain my positive evaluation score.
> >
> > Regarding the additional numerical experiments with tabular data, I acknowledge that the proposed method effectively improved the prediction performance by generating counterfactual examples.
> > While this does not fundamentally impact the core finding, I am still wondering whether the proposed method could capture the causal relationships included in the tabular data.
> >
> > Specifically, when a single categorical variable is encoded into multiple binary features, only one of the resulting binary features can have a value of 1. Therefore, if a counterfactual example requires changing one of these binary features from 0 to 1, all other binary features in that group must be changed to 0 simultaneously.  It would be helpful if the authors could clarify whether the generated counterfactual examples satisfied such a relationship.

---

> > > ### Author Response · Authors · 2025-08-06
> > >
> > > Dear Reviewer,
> > >
> > > Thank you for your insightful comments regarding the preservation of causal relationships in counterfactual generation, particularly in the context of one-hot encoded categorical features. We address the two key aspects of the causal relationship raised:
> > > 1. Correct Handling of One-Hot Encoded Categorical Variables
> > >
> > >  In our work, we employ the Tabular Variational Autoencoder (TVAE) [1] to ensure categorical data are appropriately processed. TVAE handles categorical variables by applying a Gumbel-Softmax function [2] during the decoding phase to approximate discrete outputs. Specifically, after decoding from the latent space (which is continuous and smooth by design), the output logits for each one-hot group are passed through the Gumbel-Softmax. This function amplifies the largest value in the group and suppresses the others, effectively producing a near one-hot vector.
> > > For instance, consider a one-hot encoded categorical feature decoded as [0.2,0.6,0.2]; the application of Gumbel-Softmax will yield something close to [0.01,0.98,0.01], ensuring that only one binary variable in the group dominates. This mechanism ensures that our counterfactual examples respect the constraint that only one binary feature per one-hot group can be active (i.e., equal to 1) at any time.
> > >
> > > 2. Preservation of Causal Relationships Between Features
> > >
> > > We have also observed that the counterfactual examples generated by our method maintain meaningful causal relationships between correlated features if the generative model is capable of doing so. In the Adult dataset, for example, the relationship between marital-status and relationship is preserved in the counterfactual outputs. One illustrative case involves an original data point with the attributes {marital-status: _Divorced_, relationship: _Not-in-family_}, which is transformed into a counterfactual with {marital-status: _Married-civ-spouse_, relationship: _Husband_}. This is because of TVAE’s conditional training on specific categorical values. This approach allows the model to learn feature dependencies more effectively. It's important to note that LeapFactual does not enforce any causal structure. However, we see causality as an important direction of research and aim to include it in future works.
> > >
> > > [1] Lei Xu, Maria Skoularidou, Alfredo Cuesta-Infante, Kalyan Veeramachaneni. Modeling Tabular data using Conditional GAN. NeurIPS, 2019.
> > >
> > > [2] Eric Jang, Shixiang Gu, and Ben Poole. Categorical reparameterization with gumbel-softmax. In International Conference on Learning Representations, 2016.

---

> > > > ### Comment · Area_Chair_7f8h · 2025-08-08
> > > > **Finalize discussion**
> > > >
> > > > Dear reviewer XFr6,
> > > > in response to the questions you have raised the authors provided further explanations. Do these resolve your concerns? Can you please follow up with the authors and wrap up the discussion?
> > > > Thank you,
> > > > your AC

---

> > > > > ### Comment · Reviewer_XFr6 · 2025-08-08
> > > > >
> > > > > Thank you for your clarification. My concerns have been resolved.

---

### Official Review · Reviewer_rHeX · 2025-07-02

**Clarity:** 3
**Significance:** 4
**Originality:** 4
**Rating:** 5
**Confidence:** 3

**Summary:**

The paper introduces LeapFactual, a counterfactual explanation algorithm. LeapFactual leverages conditional flow matching to generate robust and plausible counterfactuals by separating class-related features from residual information. Authors reformulate conditioning term from I-CFM to be class related. It addresses critical limitations of existing methods, such as gradient vanishing, discontinuous latent spaces, and reliance on learned decision boundaries. The method combines the strengths of optimization-based and generation-based approaches while remaining model-agnostic, enabling use with non-differentiable systems like human-in-the-loop workflows. Key contributions include:
1. Proposing LeapFactual: A CFM-based algorithm that bridges flattened and structured latent spaces, allowing to produce realistic counterfactuals even when learned decision boundaries diverge from true ones.
2. Theoretical grounding: The CE-CFM objective is theoretically justified, showing flow matching inherently disentangles class-related and residual information.
3. Empirical validation: Extensive experiments on benchmark datasets: Morpho-MNIST, Galaxy10 DECaLS and real-world tasks demonstrate superior accuracy, interpretability, and utility in enhancing model performance via robust CEs used as training data.

**Questions:**

1. Authors compare their proposed method against defined baselines, but not against other published methods. Could you compare your method against other existing methods e.g. https://arxiv.org/pdf/2210.11841
2. Could you explain why different metrics were used for different datasets? Additionally could you provide explanations on why different datasets were used to evaluate different aspects of the proposed method and not all 3 datasets for all aspects?
3. How does computational cost scale with input resolution (high-res images like FFHQ)? A brief analysis of training/inference time trade-offs compared to baselines would strengthen claims of generalization.
4. Authors mention non-differentiable models (human annotators). What safeguards prevent biases in such scenarios, and how does the method mitigate risks of propagating human errors into counterfactuals?

**Ethical Concerns:**

["NO or VERY MINOR ethics concerns only"]

**Final Justification:**

See the response to the authors.

**Limitations:**

The authors acknowledge scalability issues in high-dimensional spaces but could expand on potential mitigation strategies via OT-CFM as mentioned in future work. Ethical impacts are briefly addressed, but further analysis of bias amplification or misuse scenarios would improve robustness. Overall, the limitations are candid and well-articulated.

**Quality:**

3

**Strengths And Weaknesses:**

## Quality
### Strength
  1. The theoretical foundation, including *Theorem 1* on information compression via class conditioning, provides clear justification for the design of their CE-CFM training objective.
  2. Empirical validation is thorough: experiments on Morpho-MNIST, Galaxy 10 DECaLS, and FFHQ datasets demonstrate significant improvements over baseline Opt-based and CGM methods in terms of accuracy, AUC, and perceptual metrics like SSIM/PSNR.
  3. The method handles non-differentiable models (human-in-the-loop systems via CLIP).
  4. Robust CEs improve model training accuracy when used as augmentations, addressing real-world dataset imbalance challenges.
### Weaknesses:
- Scalability limitations in high-dimensional spaces are mentioned as a future work, but no ablation studies on computational efficiency are provided.
- While the authors discuss discrepancies between true/learned decision boundaries, they do not directly evaluate how often their method "rescues" misaligned decisions. This could be quantified with additional experiments.
- Edge Cases Unexplored: Potential failure scenarios, such as noisy inputs or extreme class imbalances beyond tested datasets, are not addressed in experiments.
---
## Clarity
### Strengths:
  - The paper is well-organized and clearly structured. Figures effectively convey complex ideas. Figure 1 represents conceptual illustration of decision boundaries and Figure 4 shows flow transports.
### Weaknesses
- The authors incorrectly use the term 'robust' when describing counterfactual explanations that lie closer to the data manifold. According to [1], this property is more accurately termed 'plausibility.' In the established literature, robustness specifically refers to the stability of counterfactual explanations under small perturbations to either the input data or the underlying model [2, 3], which is conceptually distinct from the notion of plausibility.
- Authors state that OPT-based methods face gradient vanishing problem, but this claim is unsupported.
- Authors discuss structural and flattened representations, but the definition is unclear.
- Section 3.2.1’s discussion of conditional flow matching requires deeper familiarity with I-CFM, leaving some gaps for non-experts in flow-based models.
- Algorithm 1 would benefit from pseudocode details on how `BLENDINGLEAP` and `INJECTIONLEAP` are implemented.

[1] Guidotti, Riccardo. "Counterfactual explanations and how to find them: literature review and benchmarking." _Data Mining and Knowledge Discovery_ 38.5 (2024): 2770-2824.

[2] Artelt, André, et al. "Evaluating robustness of counterfactual explanations." _2021 IEEE symposium series on computational intelligence (SSCI)_. IEEE, 2021.

[3] Jiang, Junqi, et al. "Formalising the robustness of counterfactual explanations for neural networks." _Proceedings of the AAAI conference on artificial intelligence_. Vol. 37. No. 12. 2023.

---
## Significance
### Strengths
  - The work addresses a critical challenge in counterfactual explanations - generating robust samples within distribution while handling model discrepancies.
  - Model-agnosticism enables broader adoption, including scenarios where retraining generative models per classifier is impractical.
### Weaknesses
  - No

---
## Originality
### Strengths
- The integration of *conditional flow matching* for counterfactual generation is novel.
- Combines structured latent spaces with continuous exploration.
- Introduces distinct transport operations (lifting and landing) to decouple class-related vs. residual information, enabling blending/injection beyond simple replacement.
### Weaknesses
- No

---

> ### Author Rebuttal · Authors · 2025-07-29
>
> Dear Reviewer,
>
> We sincerely thank you for your feedback and time. We appreciate the positive feedback regarding significance and originality and the helpful constructive criticism, that we think in most cases can be integrated easily. To address your concerns, we conducted an extra experiment on the **tabular adult dataset** and appended it at the end. In the following, we address each concern and question point by point.
>
> ## Quality weaknesses:
>
> **W1 (scalability limitations - no ablation on computational efficiency):**  We included reference training time in Appendix B.1.2, B.1.3, and B.1.4: The model for the MorphoMNIST, Galaxy, and FFHQ experiments takes about 80 secs, 2 hours, and 22 hours to train on a single A100 GPU, respectively. However, we emphasize that the bottleneck for the FFHQ experiment comes primarily from StyleGAN, which is a heavy generative model, taking relatively long to decode the latent vectors. The 1D-UNet model can also be improved to increase efficiency. We mention scalability limitations for high-dimensional spaces as future work, especially regarding diffusion models and normalizing flows, since the latent-space dimensions are as high as input images. Therefore, the training time is expected to be much longer.
>
> **W2 (rescue misaligned decisions - additional experiments):** We address differences in true and learned decision boundaries as stated, however directly measuring how often we ‘rescue’ samples due to misalignment seems not trivial. As an indirect measurement, we refer to the experiment in Appendix B.1.3 (Ablation Study) and the new adult experiment (please see below). The experiments show that the accuracy of the **strong** classifier increases when we increase the number of injection steps, indicating that the performance on rescuing samples from misalignment depends on the number of injection steps.
>
> **W3 (potential failure scenarios like noisy input or class imbalances):** We begin with addressing the edge case of noisy samples, which we understand as noisy data rather than noisy labels, unlike in Q4. If the generative model provided by the user is robust to noise, the latent space itself will be more or less the same. Since we apply our method on the latent space, our approach depends on the robustness of the generative model. Secondly, the new adult tabular data experiment shows that the method is robust to imbalanced data (20% high income vs. 80% low income) at least to some degree. More experiments are needed to measure this robustness, as mentioned in our future work section. We emphasize that common upsampling or downsampling strategies can be applied here to rescue the imbalanced data problem.
>
> ## Clarity weaknesses:
>
> **W1 (incorrectly named robust instead of plausible):** Thank you for pointing this out. We agree that our definition of robustness is not favorable since it will lead to confusion. We want to stress the fact that the produced counterfactuals are dependable, thus plausibility does not really capture what we want to express. However, we agree with you that renaming is necessary, and therefore, we will update the notation in the camera-ready version to **reliable counterfactuals**.
>
> **W2 (OPT gradient vanishing - claim unsupported):** We understand that the claim seems to be unsupported. The gradient vanishing can be explained by the theory behind optimization-based methods. If we look at the loss function (Equation 1), we can find that it depends on the learned decision boundary, and the first summand becomes zero as soon as it reaches the “correct” class (regarding the learned decision boundary). Since we assume there to be an imperfect classifier, we see this leading to gradient vanishing. This can also be observed in Appendix B.1.3 (Ablation Study) and the adult experiment, where the accuracy of counterfactuals generated by the **weak classifier** is very low regarding the **strong classifier**.
>
> **W3 (structural and flattened representations, definition unclear)+ W4 (discussion of conditional flow matching) + W5 (pseudocode details for Blendingleap and Injectionleap):** Thank you for the hints, we will improve the definition of structural and flattened representation to be more precise and elaborate on conditional flow matching in the appendix of the camera-ready version. Similarly, the pseudocode will also be integrated in the appendix.
>
> ## Questions:
>
> **Q1 (Compare with other baselines):** We focus on a fair comparison and therefore we want to compare against competitors with the same starting points, which are the models in our case. For all approaches, we compare against the paradigms to showcase the potential of our approach, rather than comparing specific models that belong to one of the approaches. To our knowledge, the referred work uses a classifier-guided diffusion model to generate CE, which is in the optimization-based paradigm. However, because generalizing to diffusion models would be our future work, we really appreciate the reviewer’s suggestion and will carefully read the paper.
>
> **Q2 (Different metrics and datasets):** The MorphoMNIST dataset is a specifically designed benchmark dataset that provides reliable metrics within its implementation. Unfortunately, these metrics can not be easily transferred to other datasets. The classifier accuracy for MorphoMNIST is already high, therefore the model improvement experiment for this dataset can not provide any additional information. For FFHQ, there is no ground truth label on whether the person is smiling or not; therefore, we can not do the model improvement experiment. For the generalization experiment, the MorphoMNIST and Galaxy data are trivial (or have been demonstrated to work), and we need a more challenging dataset.
>
> **Q3 (Computation cost):** We did include training time for the experiments in Appendix B.1.2 to B.1.4. The inference time is in the scale of seconds or minutes, depending on the problem difficulty. See also W1.
>
> **Q4 (Human error):** Thank you for bringing up this discussion. We assume that misalignment will always happen. Human annotation errors cause decision boundary misalignment. Our reliable counterfactuals mitigate this, as they do not strictly rely on the learned boundary. The Adult experiment shows the approach works even when labels are intentionally flipped.
>
> ## Extra experiment
> We conducted an additional experiment using the Adult dataset [1]. The dataset was split into training and test sets using an 80/20 ratio.
>
> We trained a weak classifier with label noise, achieved an accuracy of 0.7093, and an AUC of 0.6874. The strong classifier trained on clean data reached 0.8546 accuracy and 0.9039 AUC.
>
> We then trained a TVAE [2] generative model and a flow matching model conditioned on the weak classifier's predictions to generate counterfactuals (CE) and reliable CE (target labels = flipped predictions). Both classifiers evaluated the generated samples.
>
> To measure the similarity, Categorical features were one-hot encoded, and numerical features were normalized to enable the Mean Squared Error (MSE) metric.
>
> From the results, we observe the following:
> - **Weak classifier's** accuracy increases with blending steps but drops sharply during injection.
> - **Strong classifier's** accuracy starts below the weak classifier's during blending but surges from ~73% to ~93% after 10 injection steps, reaching 97% in the end.
>
> These findings demonstrate that:
> - The proposed method is effective on tabular data, provided the generative model is capable of capturing the underlying causal relationships.
> - The method is robust to imbalanced datasets (the Adult dataset contains 80% low-income and 20% high-income labels) and can handle noisy data (e.g., human annotation errors), as evidenced by the performance of the weak classifier trained with noisy labels.
> - The reliable counterfactual examples effectively resolve the misalignment between the learned and true decision boundaries, under the assumption that the strong classifier better approximates the true decision boundary than the weak classifier.
>
> | Metric        | N_b=30         | N_b=60         | N_b=90         | N_i=10         | N_i=20         | N_i=30         |
> |---------------|-------------------------|-------------------------|-------------------------|-------------------------|-------------------------|-------------------------|
> | **ACC↑**     | 0.7607 ± 0.0027         | 0.8355 ± 0.0026         | **0.8503 ± 0.0026**     | 0.6984 ± 0.0016         | 0.6686 ± 0.0020         | 0.6389 ± 0.0013         |
> | **AUC↑**     | 0.8522 ± 0.0014         | 0.8989 ± 0.0017         | **0.9082 ± 0.0016**     | 0.8070 ± 0.0011         | 0.7905 ± 0.0015         | 0.7713 ± 0.0011         |
> | **ACC_r↑** | 0.7232 ± 0.0017         | 0.7258 ± 0.0017         | 0.7276 ± 0.0019         | 0.9328 ± 0.0016         | 0.9655 ± 0.0009         | **0.9692 ± 0.0009**     |
> | **AUC_r↑** | 0.7841 ± 0.0019         | 0.7860 ± 0.0018         | 0.7874 ± 0.0021         | 0.9466 ± 0.0019         | 0.9747 ± 0.0011         | **0.9783 ± 0.0011**     |
> | **MSE↓**     | **0.0537 ± 0.0001**     | 0.0544 ± 0.0001         | 0.0546 ± 0.0001         | 0.0688 ± 0.0002         | 0.0717 ± 0.0002         | 0.0731 ± 0.0002         |
>
> *Metrics with mean ± standard error for 5 runs. Bold values represent the best-performing configuration for each metric. The first three columns (N_b) indicate performance changes across blending steps, while the last three columns (N_i) indicate performance changes during injection steps, which continue from the last blending step. $\gamma_b = 0.1$, $\gamma_{i,lift}=1$, and $\gamma_{i,land}=1.1$. ACC and AUC are evaluated using the weak classifier, whereas ACC_r and AUC_r are evaluated using the strong classifier.*
>
> [1] Becker, B. & Kohavi, R. (1996). Adult [Dataset]. UCI Machine Learning Repository.
>
> [2] Lei Xu, Maria Skoularidou, Alfredo Cuesta-Infante, Kalyan Veeramachaneni. Modeling Tabular data using Conditional GAN. NeurIPS, 2019.

---

> > ### Comment · Reviewer_rHeX · 2025-08-05
> >
> > I appreciate the thorough feedback and additional experiment you provided. The paper meets my expectations, and I will maintain my current evaluation rating.

---

> > > ### Author Response · Authors · 2025-08-06
> > >
> > > Dear Reviewer,
> > >
> > > Thank you for your time and acknowledging our effort!

---

### Decision · Program_Chairs · 2025-09-17

**Decision:**

Accept (poster)

**Comment:**

a) summary: The paper introduces a novel method for generating counterfactual explanations addressing the shortcomings of optimization CE methods (vanishing gradients) and other conditional generative methods (discontinuity of latent space). It modyfies the initial I-CFM formulation from [42] to be class conditioned and use it to remove and re-introduce class relevant information to the data through blending and optionally injection leaps (lifting and landing transports). The experiments explore the effectiveness of the method both quantiatively and qualitatively and show superior performance with respect to baselines.

(b) strengths: Technically strong paper with solid theoretical background. Convincing experimental evaluation. Flexibility of the method to operate even over non-differentiable classifiers including human-in-the-loop systems.

(c) weaknesses: Clarity of some statements and used terminology, high computational costs and bad scalability to high-dimensional spaces.

(d) reasons for accept: Explainability is an important topic with CE an increasingly popular approach. The paper takes a fresh look that is theoretically well grounded while addressing clearly stated shortcomings of previous methods. The experimental evaluation looks convincing.

(e) review discussion: The reviewers appreciated the novelty and good theoretical grounding of the method with clear justification for the selected approach. They expressed some concerns about clarity of some statements/terms (e.g. robustness of CE, structural/flattened representation), missing support for some claims (e.g. vanishing gradients), computational costs and scalability, practical implementation details and hyperparameter setting. The authors addressed most of these very clearly and open within the rebuttal often by referring to existing appendix, added an experiment on tabular data, and acknowledged some existing limitations (mainly scalability to high-dimensional spaces) left for future work. The reviewers generally appreciated the answers and regarded the discussion as useful with positive outcomes.